# Safety outcomes following COVID-19 vaccination and infection in 5.1 million children in England

Emma Copland [1], Martina Patone[1], Defne Saatci [1], Lahiru Handunnetthi[2,3], Jennifer Hirst[1], David P. J. Hunt [4], Nicholas L. Mills [5,6], Paul Moss [7], Aziz Sheikh [1,6], Carol A. C. Coupland[1,8], Anthony Harnden[1], Chris Robertson [9] & Julia Hippisley-Cox [1] ✉

The risk-benefit profile of COVID-19 vaccination in children remains uncertain. A self-controlled case-series study was conducted using linked data of 5.1 million children in England to compare risks of hospitalisation from vaccine safety outcomes after COVID-19 vaccination and infection. In 5-11-year-olds, we found no increased risks of adverse events 1–42 days following vaccination with BNT162b2, mRNA-1273 or ChAdOX1. In 12-17-year-olds, we estimated 3 (95%CI 0–5) and 5 (95%CI 3–6) additional cases of myocarditis per million following a first and second dose with BNT162b2, respectively. An additional 12 (95%CI 0–23) hospitalisations with epilepsy and 4 (95%CI 0–6) with demyelinating disease (in females only, mainly optic neuritis) were estimated per million following a second dose with BNT162b2. SARS-CoV-2 infection was associated with increased risks of hospitalisation from seven outcomes including multisystem inflammatory syndrome and myocarditis, but these risks were largely absent in those vaccinated prior to infection. We report a favourable safety profile of COVID-19 vaccination in under-18s.

The United Kingdom (UK) approved COVID-19 vaccination for all children aged 12 and over in September 2021. This was extended to 5–11-year-olds in April 2022 in a one-off programme offering a primary course of COVID-19 vaccination to children who were not in a vulnerable or high-risk group[1]. Despite uptake being very high in adults, with over 80% receiving at least one dose of COVID-19 vaccine as of 11th May 2023, uptake has been lower in children, with 62% of 16–17-year-olds, 46% of 12–15-year-olds and 10% of 5–11-year-olds being vaccinated against COVID-19[2]. In the UK, the vast majority of vaccinated children received the BNT162b2 (Pfizer/BioNTech) COVID-19 vaccine, as the UK Joint Committee on Vaccination and Immunisation (JCVI) advised

against using ChAdOX1 (AstraZeneca) in people under 40 or mRNA-1273 (Moderna/SpikeVax) in children under 18 during the time period that most children were vaccinated[3]. Those aged 12 years and above were given a full dose of BNT162b2 vaccine (30 micrograms), and children aged 5-11 years were given a dose of 10 micrograms of BNT162b2 vaccine[3].

In November 2022, the JCVI recommended that 16–49-year-olds who are not in a clinical risk group should no longer be offered a third dose of vaccine from February 2023, and that primary course vaccination in 5–49-year-olds should be targeted to groups at high risk of severe COVID-19[4]. They have advised to continue vaccinating clinically

[1]Nuffield Department of Primary Health Care Sciences, University of Oxford, Oxford, UK. [2]Centre for Human Genetics, University of Oxford, Oxford, UK. [3]Department of Psychiatry, University of Oxford, Oxford, UK. [4]UK Dementia Research Institute, Centre for Clinical Brain Sciences, University of Edinburgh, Edinburgh, UK. [5]BHF/University Centre for Cardiovascular Science, University of Edinburgh, Edinburgh, UK. [6]Usher Institute, University of Edinburgh, Edinburgh, UK. [7]Institute of Immunology and Immunotherapy, University of Birmingham, Birmingham, UK. [8]Centre for Academic Primary Care, School of Medicine, University of Nottingham, Nottingham, UK. [9]Department of Mathematics and Statistics, University of Strathclyde, Glasgow, UK. ✉ e-mail: julia.hippisley-cox@phc.ox.ac.uk

vulnerable children aged 6 months and above and to vaccinate otherwise healthy children aged 12 years and above living with immunosuppressed individuals[4].

Although the benefits of COVID-19 vaccines in older adults clearly outweigh the risks of rare complications[5–8], the balance of risks and benefits in young people remains uncertain. Whilst clinical trials have demonstrated the effectiveness of COVID-19 vaccines in reducing the risk of severe COVID-19 in children aged 5–15 years, the absolute risk of severe outcomes, including hospitalisation, intensive care unit (ICU) admission and death, following infection is low[9–12]. A serious consequence of SARS-CoV-2 infection in children is multisystem inflammatory syndrome (MIS-C)[13,14], which can lead to coronary artery aneurysms, cardiac dysfunction, and multiorgan inflammatory manifestations[15]. Post-COVID syndrome, or long COVID, is another serious outcome of SARS-CoV-2 infection[16]. The possible impact of long COVID in children is still unclear, but it is estimated to affect 10% of those infected by SARS-CoV-2 and could potentially result in lifelong illness[16,17]. COVID-19 vaccines in childhood may reduce the risk of MIS-C[13] and long COVID[17], and secondary benefits might include increasing overall population immunity, thereby minimising disruptions to education and maintaining overall well-being and health within this age group.

Concerns around vaccine safety have been identified as a barrier to COVID-19 vaccine acceptance, particularly in the context of parents and guardians giving consent for their children to be vaccinated[18–20]. An increased risk of myocarditis has been consistently reported after the delivery of mRNA vaccines, predominantly affecting males aged 18 to 35 years and most notably after the second dose[21,22]. This elevated risk of mRNA vaccination-associated myocarditis has also been reported in adolescent males, although these cases have generally been mild and the long-term prognosis is favourable[23–32]. No serious safety concerns have yet been raised in younger children; however, population-based studies assessing the risk of adverse events following vaccination compared with an unvaccinated group are lacking[33–35].

In light of the JCVI's advice to continue vaccinating specific groups of children against COVID-19, and the potential for future mass vaccination programmes if new SARS-CoV-2 variants emerge or the number of severe COVID-19 cases increase, it remains important to quantify the overall risks and benefits of COVID-19 vaccination in this age group to inform future vaccine policy. Therefore, our primary aim was to investigate and compare the risks of pre-specified vaccine safety outcomes following vaccination with BNT162b2, mRNA-1273 and ChAdOX1 in children. We also aimed to compare the risks of these safety outcomes following SARS-CoV-2 infection in vaccinated and unvaccinated children, as a guide to inform global public health policy considerations.

We used the English National Immunisation Management Service (NIMS) database of COVID-19 vaccination, linked at the individual-level to national data for mortality, hospital admissions, and SARS-CoV-2 infection. We undertook a self-controlled case series design, originally developed to examine vaccine safety[36,37], to investigate the association between COVID-19 vaccines available in the UK between 8th December 2020 and 7th August 2022 (BNT162b2, mRNA-1273 and ChAdOx1) and hospitalisation with the following pre-specified outcomes: myocarditis[21,22], MIS-C[38], immune thrombocytopenia (ITP)[39], epilepsy[40], acute pancreatitis[41], acute disseminated encephalomyelitis (ADEM)[42], Guillain-Barre syndrome[43], appendicitis[44], demyelinating disease[6], myositis[45], angioedema[46] and anaphylaxis[46]. We also investigated the association of SARS-CoV-2 infection with these outcomes in children who had been vaccinated prior to infection compared to those who were unvaccinated at time of infection. We compared the incidence of hospitalisation from each outcome in the six weeks following vaccination or SARS-CoV-2 infection relative to the baseline period, and estimated the absolute risk as the excess number of events expected per million children exposed. We also conducted a matched cohort analysis using vaccinated and unvaccinated children included the QResearch primary care database to improve the robustness of the study.

## Results

### Number of children receiving COVID-19 vaccines
A total of 5,197,925 young people aged 5–17 years, comprising 1,842,159 children aged 5–11 years and 3,355,766 adolescents aged 12–17 years were included in the study. 4,347,781 young adults aged 18–24 years, were included as a comparison. The characteristics of the young people included in the study are detailed in Table 1 and Supplementary Table 1.

In children aged 5–11 years, 32% (n = 581,545) received at least one dose of COVID-19 vaccine and 16% (n = 303,118) received a second dose within the study period. Over 99.9% of 5–11-year-olds who received at least one COVID-19 vaccine dose received the BNT162b2 vaccine (Table 2). 82% (n = 1,508,661) of children in this age group had a positive SARS-CoV-2 test recorded between 8th December 2020 and 7th August 2022, 0.4% (n = 5665) of whom received their first COVID-19 vaccine dose prior to their first recorded positive SARS-CoV-2 test (Table 2).

In adolescents aged 12–17 years, 86% (n = 2,882,229) received a first dose of COVID-19 vaccine, 67% (n = 2,254,214) received a second dose and 14% (n = 454,868) received a third dose (Table 2).

The characteristics of the population excluded from the self-controlled case series analysis (i.e. those who did not receive any COVID-19 vaccine and did not have a positive SARS-CoV-2 test recorded during the study period) are presented in Supplementary Table 2.

### Risk of pre-specified safety outcomes following COVID-19 vaccination in children aged 5–11 years
In children aged 5–11 years, we did not observe an increased risk of any of the pre-specified outcomes in the 1–42 days following any dose COVID-19 vaccine with BNT162b2, mRNA-1273 or ChAdOX1 (Table 3). However, given that less than 0.1% of vaccinated 5-11-year-olds received a ChAdOX1 or mRNA-1273 vaccine, the probability of type II errors was high as the sample size was too small to detect statistically significant associations for these vaccines.

The clinical characteristics of all children hospitalised with a pre-specified safety event are shown in Supplementary Table 3. Supplementary Tables 4–7 show the incidence rate ratios (IRR) and 95% confidence intervals (CI) for all outcomes 1–42 days and in weekly risk periods following each vaccine dose in 5–11-year-olds in males and females separately, and the effect of ethnicity on the risk of each outcome.

### Risks of pre-specified safety outcomes following COVID-19 vaccination in adolescents aged 12–17 years
In the 1–42 days after the first and second doses of BNT162b2, we observed an increased risk of myocarditis in adolescents aged 12–17 years (IRR 1.92, 95%CI 1.08–3.43 and IRR 2.96, 95%CI 1.65–5.32 for first and second dose, respectively) (Table 4). We estimated that an additional 3 (95%CI 0–5) cases per million exposed would be anticipated after the first dose and 5 (95%CI 3–6) after the second dose (Fig. 1). When we split the risk period into weekly blocks, the increased risk was restricted to 1–14 days following each dose (Supplementary Table 8). There was also an increased risk of hospitalisation with epilepsy (IRR 1.17, 95%CI 1.00–1.37; excess events per million: 12, 95%CI 0–23) in the 1–42 days following the second dose of BNT162b2 (Table 4, Fig. 1).

In the sex-stratified analysis, the increased risk of myocarditis after the first dose of BNT162b2 was only observed in females (IRR 4.01, 95%CI 1.33–12.09; excess events per million: 3, 95%CI 1–4), while the increased risk following the second dose was observed in males only (IRR 2.87, 95%CI 1.50–5.51; excess events per million: 9, 95%CI 4–11) (Supplementary Figs. 1 & 2, Supplementary Tables 4 & 5). We additionally observed an increased risk of demyelinating disease, restricted to females (IRR 2.41, 95%CI 1.06-5.48; excess events per million: 4, 95%CI 0–6) following the second dose of BNT162b2. Of the eight female adolescents who experienced demyelinating disease in

**Table 1 | Baseline demographic characteristics at beginning of study period of cohort in children aged 5–11 and 12–17 years who received at least one COVID-19 vaccine dose or tested positive for SARS-CoV-2 in England between 8th December 2020 and 7th August 2022**

| | 5–11 years (n = 1,842,159) | | 12–17 years (n = 3,355,766) | |
| --- | --- | --- | --- | --- |
| | Received at least one COVID-19 vaccine | Positive SARS-CoV-2 test | Received at least one COVID-19 vaccine | Positive SARS-CoV-2 test |
| Total N (%) | 581,545 (31.6) | 1,508,661 (81.9) | 2,882,229 (85.9) | 1,602,710 (47.8) |
| Age, mean (SD) | 8.7 (2.0) | 8.4 (1.9) | 14.6 (1.8) | 14.3 (1.7) |
| Sex | | | | |
| Female | 45.3 (263,552) | 47.5 (716,227) | 47.1 (1,356,342) | 49.5 (793,332) |
| Male | 47.9 (278,675) | 48.6 (733,779) | 47.0 (1,355,952) | 45.8 (733,890) |
| Not recorded | 6.8 (39,318) | 3.9 (58,655) | 5.9 (169,935) | 4.7 (75,488) |
| Ethnicity | | | | |
| White | 69.0 (401,542) | 75.8 (1,143,151) | 69.5 (2,001,849) | 72.4 (1,161,127) |
| Indian | 5.2 (30,202) | 2.7 (40,444) | 2.8 (80,872) | 2.2 (36,026) |
| Pakistani | 1.7 (10,001) | 2.0 (30,619) | 2.4 (69,239) | 2.1 (33,269) |
| Bangladeshi | 1.0 (5705) | 0.6 (9573) | 1.0 (27,390) | 0.7 (11,199) |
| Other Asian | 2.2 (12,549) | 1.3 (19,912) | 1.6 (46,241) | 1.3 (20,755) |
| Black Caribbean | 0.1 (821) | 0.5 (6822) | 0.3 (9217) | 0.6 (9386) |
| Black African | 1.2 (7176) | 1.3 (19,726) | 1.5 (44,384) | 1.5 (24,089) |
| Chinese | 0.9 (5287) | 0.4 (5939) | 0.5 (13,844) | 0.3 (4892) |
| Other | 4.8 (27,856) | 5.1 (77,488) | 3.9 (111,050) | 4.2 (67,450) |
| Not recorded | 13.8 (80,406) | 10.3 (154,987) | 16.6 (478,143) | 14.6 (234,517) |
| Medical history* | | | | |
| Guillain-Barre syndrome | (< 5) | < 0.1 (10) | < 0.1 (35) | < 0.1 (25) |
| Appendicitis | 0.1 (698) | 0.1 (1929) | 0.2 (7030) | 0.3 (4141) |
| Demyelinating disease | < 0.1 (28) | < 0.1 (66) | < 0.1 (162) | < 0.1 (82) |
| Myositis | < 0.1 (40) | < 0.1 (86) | < 0.1 (143) | < 0.1 (68) |
| Myocarditis | < 0.1 (5) | (< 5) | < 0.1 (35) | < 0.1 (15) |
| Acute pancreatitis | < 0.1 (20) | < 0.1 (46) | < 0.1 (220) | < 0.1 (121) |
| Immune thrombocytopenia | < 0.1 (237) | < 0.1 (571) | < 0.1 (725) | < 0.1 (443) |
| Multisystem inflammatory syndrome | < 0.1 (53) | < 0.1 (120) | < 0.1 (76) | < 0.1 (36) |
| Anaphylaxis | < 0.1 (88) | < 0.1 (269) | < 0.1 (478) | < 0.1 (314) |
| Epilepsy | 0.4 (2062) | 0.2 (3038) | 0.2 (5746) | 0.2 (2644) |
| Angioedema | < 0.1 (96) | < 0.1 (241) | < 0.1 (368) | < 0.1 (239) |
| Acute disseminated encephalomyelitis | < 0.1 (53) | < 0.1 (118) | < 0.1 (159) | < 0.1 (80) |

*Defined as hospital admission with ICD-10 code for condition in two years prior to study start date.
Cells with < 5 are suppressed. Data are presented as column % (counts).

the 1–42 days following a second dose of BNT162b2, five were coded as optic neuritis.

In a *post hoc* analysis investigating differences in risk between children of different ethnic backgrounds, we found that the risk of anaphylaxis following a second dose of BNT162b2 in adolescents with non-white ethnicity was higher relative to the risk in adolescents with white ethnicity (relative IRR 2.55, 95%CI 1.00–6.46) (Supplementary Table 6). However, when the analysis was restricted to the subgroup of adolescents from non-white ethnic backgrounds, the risk of anaphylaxis in the 1–42 days following a second dose of BNT162b2 was not significantly increased compared to the baseline period (IRR 1.69, 95% CI 0.80–3.54) (Supplementary Table 6). We did not identify any differences in vaccine safety between white and non-white ethnicity for any of the other pre-specified outcomes in under-18s.

We found no evidence for significantly increased risks for any of the pre-specified outcomes in the 1–42 days following a first, second or third dose of mRNA-1273 vaccine in 12–17-year-olds (Table 4). However, this analysis lacked power to detect statistically significant associations, except for very large effect sizes, as less than 0.1% of adolescents received a first or second dose of mRNA-1273 vaccine.

There was an increased risk of hospitalisation with epilepsy 1–42 days after a first dose of ChAdOX1 (IRR 1.93, 95%CI 1.10–3.39), with an additional 705 (95%CI 129–1033) cases expected per million exposed (Table 4). This increased risk was restricted to females in the sex-stratified analysis (IRR 2.26, 95%CI 1.03–4.94) with an additional 813 (95%CI 44–1164) hospitalisations with epilepsy expected per million female adolescents exposed (Supplementary Table 4).

We also observed an increased risk of appendicitis in the 1–42 days following the second dose of ChAdOX1 (IRR 4.64, 95%CI 1.77–12.17; excess events per million: 512, 95%CI 283–599) (Table 4).

The IRRs and 95% CIs for all outcomes 1–42 days and in weekly risk periods following each vaccine dose in 12–17-year-olds in males and females separately, and the effect of ethnicity on the risk of each outcome, are presented in Supplementary Figs. 1 & 2, Supplementary Tables 4–6 & 8.

### Risks of pre-specified safety outcomes following COVID-19 infection in children aged 5–11 years

In children aged 5-11 years who had received at least one dose of COVID-19 vaccine before the date that their positive SARS-CoV-2 test

**Table 2 | Characteristics of COVID-19 vaccination and/or infection in children aged 5-11 and 12-17 years who received at least one COVID-19 vaccine dose or tested positive for SARS-CoV-2 in England between 8th December 2020 and 7th August 2022**

| | 5-11 years (*n* = 1,842,159) | | | | 12-17 years (*n* = 3,355,766) | | | |
|---|---|---|---|---|---|---|---|---|
| | Dose 1 | Dose 2 | Dose 3 | Positive SARS-CoV-2 test | Dose 1 | Dose 2 | Dose 3 | Positive SARS-CoV-2 test |
| Total N | 581,545 | 303,118 | 870 | 1,508,661 | 2,882,229 | 2,254,214 | 454,868 | 1,602,710 |
| Type of vaccine | | | | | | | | |
| BNT162b2 | 581,356 (> 99.9) | 303,045 (> 99.9) | 870 (100) | - | 2,870,403 (99.6) | 2,240,546 (99.4) | 399,845 (87.9) | - |
| ChAdOX1 | 97 (< 0.1) | 13 (< 0.1) | 0 (0) | - | 10,245 (0.4) | 9190 (0.4) | 46 (< 0.1) | - |
| mRNA-1273 | 92 (< 0.1) | 60 (< 0.1) | 0 (0) | - | 1581 (< 0.1) | 4478 (0.2) | 54,977 (12.1) | - |
| Vaccine status | | | | | | | | |
| No vaccine | - | - | - | 83.6 (1,260,614) | - | - | - | 29.5 (473,537) |
| Dose 1 | 100 (581,545) | 100 (303,118) | 100 (870) | 16.4 (248047) | 100 (2,882,229) | 100 (2,254,214) | 100 (454,868) | 70.5 (1,129,173) |
| Dose 2 | 52.1 (303,118) | 100 (303,118) | 100 (870) | 8.9 (133998) | 78.2 (2,254,214) | 100 (2,254,214) | 100 (454,868) | 54.4 (872,128) |
| Dose 3 | 0.1 (870) | 0.3 (870) | 100 (870) | < 0.1 (431) | 15.8 (454,868) | 20.2 (454,868) | 100 (454,868) | 9.9 (159,350) |
| Positive SARS-CoV-2 test during study period | | | | | | | | |
| None | 57.3 (333,498) | 55.8 (169,120) | 50.5 (439) | - | 60.8 (1,753,056) | 61.3 (1,382,086) | 65.0 (295,518) | - |
| Prior to first vaccine dose | 41.7 (242,382) | 43.2 (131,034) | 40.8 (355) | 99.6 (1,502,996) | 20.8 (598,477) | 20.1 (454,034) | 12.0 (54,765) | 66.9 (1,072,014) |
| After first vaccine dose | 1.0 (5665) | 1.0 (2964) | 8.7 (76) | 0.4 (5665) | 18.4 (530,696) | 18.5 (418,094) | 23.0 (104,585) | 33.1 (530,696) |

Cells with < 5 are suppressed. Data are presented as column % (counts).

was recorded, we did not observe increased risks of any of the pre-specified outcomes in the 1–42 days following SARS-CoV-2 infection. In children who had not been vaccinated against COVID-19 prior to infection, there was an increased risk of hospital admission with MIS-C following a SARS-CoV-2 positive test (IRR 11.52, 95%CI 9.25–14.36), with an additional 137 (95%CI 134–140) cases expected per million exposed (Table 3, Fig. 2). In the sex-stratified analysis, the risk of MIS-C was slightly greater in male children (IRR 12.00, 95%CI 8.92–16.12; excess events per million: 162, 95%CI 157-166) compared to female children (IRR 11.13, 95%CI 7.96–15.57; excess events per million: 124, 95%CI 119–127) (Supplementary Figs. 3 & 4, Supplementary Tables 4 & 5). The increased risk was mainly observed in the 22–42 days following the date that the positive SARS-CoV-2 test was recorded (Supplementary Table 7).

We also observed increased risks of hospital admission for myositis, myocarditis, acute pancreatitis and ADEM following SARS-CoV-2 infection before vaccination (Table 3, Fig. 2). In the sex-stratified analyses, we additionally identified increased risks of ITP (in both sexes) and anaphylaxis (in females only) (Supplementary Figs. 3 & 4, Supplementary Tables 4 & 5).

The IRRs and 95% CIs for all outcomes 1–42 days following SARS-CoV-2 infection in 5-11-year-olds in males and females separately, and the effect of ethnicity on the risk of each outcome, are presented in Supplementary Figs. 3 & 4, Supplementary Tables 4, 5, 6 & 7.

**Risks of pre-specified safety outcomes following COVID-19 infection in adolescents aged 12–17 years**

In adolescents aged 12–17 years, we observed an increased risk of MIS-C (IRR 12.38, 95%CI 8.88–17.28; excess events per million: 84, 95%CI 81–86) in the 1–42 days following a SARS-CoV-2 infection in those who had not been vaccinated prior to SARS-CoV-2 infection (Table 4, Fig. 1). In the sex-stratified analysis, male adolescents were at higher risk of MIS-C following infection compared to females (IRR 12.33, 95%CI 8.31–18.31 and IRR 13.11, 95%CI 6.90–24.91 in males and females,

respectively), with an additional 131 (95%CI 126–135) cases expected per million males exposed compared to 48 (95%CI 44-50) in females (Supplementary Figs. 1 & 2, Supplementary Tables 4 & 5).

We also observed increased risks of hospitalisation with myocarditis, ITP and epilepsy in the 1-42 days following SARS-CoV-2 infection in adolescents who had not been vaccinated against COVID-19 prior to infection as well as an increased risk of hospitalisation with epilepsy in those who had received at least one vaccine dose prior to infection (Table 4, Fig. 1). In the sex-stratified analysis, the increased risks of hospitalisation with myocarditis and epilepsy were restricted to males while the increased risk of ITP following infection was only observed in females (Supplementary Figs. 1 & 2, Supplementary Tables 4 & 5). We additionally identified increased risks of appendicitis (in females only) and anaphylaxis (in males only) following SARS-CoV-2 infection.

The IRRs and 95% CIs for all outcomes 1–42 days following SARS-CoV-2 infection in 12-17-year-olds in males and females separately, and the effect of ethnicity on the risk of each outcome, are presented in Supplementary Figs. 1 & 2, Supplementary Tables 4, 5, 6 & 8.

The results for all analyses in young adults aged 18-24 years are presented in Supplementary Tables 6 & 9.

**Robustness of the self-controlled case series results**

The robustness of the results of the self-controlled case series analyses were assessed by (1) checking that the risk of outcomes during the pre-vaccination period (month prior to vaccination to account for potential bias of people with recent hospitalisation being less likely to get vaccinated) was lower than the baseline period and (2) checking that the risk of the positive control outcome (anaphylaxis) was higher following vaccination or SARS-CoV-2 infection than the baseline period. In the vast majority of analyses the estimates of the pre-vaccination period and the risk of anaphylaxis following vaccination or SARS-CoV-2 agreed with what was expected (Supplementary Tables 8 & 9).

**Table 3 | Incidence rate ratios and estimated number of excess events per million for individual outcomes in the 1–42 days following vaccination or a SARS-CoV-2 positive test (before or after vaccination) compared to baseline period in children aged 5-11 years**

| | BNT162b2 vaccine | | | mRNA-1273 vaccine | | | ChAdOX1 vaccine | | | Positive SARS-CoV-2 test | | |
|---|---|---|---|---|---|---|---|---|---|---|---|---|
| | N | IRR (95% CI) | Excess events (95% CI) | N | IRR (95% CI) | Excess events (95% CI) | N | IRR (95% CI) | Excess events (95% CI) | N | IRR (95% CI) | Excess events (95% CI) |
| **Multisystem inflammatory syndrome** | | | | | | | | | | | | |
| First dose/positive test (before vaccine) | < 5 | - | - | 0 | - | - | 0 | - | - | 226 | 11.52 (9.25, 14.36) | 137 (134, 140) |
| Second dose/positive test (after vaccine) | 0 | - | - | 0 | - | - | 0 | - | - | 0 | - | - |
| Third dose | 0 | - | - | 0 | - | - | 0 | - | | | | |
| **Myocarditis** | | | | | | | | | | | | |
| First dose/positive test (before vaccine) | 0 | - | - | 0 | - | - | 0 | - | - | 6 | 14.00 (3.40, 57.67) | 4 (3, 4) |
| Second dose/positive test (after vaccine) | 0 | - | - | 0 | - | - | 0 | - | - | 0 | - | - |
| Third dose | 0 | - | - | 0 | - | - | 0 | - | | | | |
| **Immune or idiopathic thrombocytopenia** | | | | | | | | | | | | |
| First dose/positive test (before vaccine) | 10 | * | - | 0 | - | - | 0 | - | - | 61 | * | - |
| Second dose/positive test (after vaccine) | <5 | * | - | 0 | - | - | 0 | - | - | 0 | * | - |
| Third dose | 0 | * | - | 0 | - | - | 0 | - | | | | |
| **Epilepsy** | | | | | | | | | | | | |
| First dose/positive test (before vaccine) | 61 | 1.00 (0.76, 1.33) | 0 | 0 | - | - | 0 | - | - | 100 | 1.13 (0.91, 1.39) | 0 |
| Second dose/positive test (after vaccine) | 27 | 0.91 (0.60, 1.37) | 0 | 0 | - | - | 0 | - | - | < 5 | - | - |
| Third dose | 0 | - | - | 0 | - | - | 0 | - | | | | |
| **Acute pancreatitis** | | | | | | | | | | | | |
| First dose/positive test (before vaccine) | < 5 | - | - | 0 | - | - | 0 | - | - | 9 | 2.69 (1.21, 5.97) | 4 (1, 5) |
| Second dose/positive test (after vaccine) | 0 | - | - | 0 | - | - | 0 | - | - | 0 | - | - |
| Third dose | 0 | - | - | 0 | - | - | 0 | - | | | | |
| **Myositis** | | | | | | | | | | | | |
| First dose/positive test (before vaccine) | < 5 | - | - | 0 | - | - | 0 | - | - | 15 | 5.07 (2.62, 9.83) | 8 (6, 9) |
| Second dose/positive test (after vaccine) | 0 | - | - | 0 | - | - | 0 | - | - | 0 | - | - |
| Third dose | 0 | - | - | 0 | - | - | 0 | - | | | | |
| **Acute disseminated encephalomyelitis** | | | | | | | | | | | | |
| First dose/positive test (before vaccine) | < 5 | - | - | 0 | - | - | 0 | - | - | 12 | 2.58 (1.32, 5.03) | 5 (2, 6) |
| Second dose/positive test (after vaccine) | 0 | - | - | 0 | - | - | 0 | - | - | 0 | - | - |
| Third dose | 0 | - | - | 0 | - | - | 0 | - | | | | |
| **Demyelinating disease** | | | | | | | | | | | | |
| First dose/positive test (before vaccine) | < 5 | - | - | 0 | - | - | 0 | - | - | 6 | 1.80 (0.72, 4.52) | 0 |
| Second dose/positive test (after vaccine) | < 5 | - | - | 0 | - | - | 0 | - | - | 0 | - | - |
| Third dose | 0 | - | - | 0 | - | - | 0 | - | | | | |
| **Appendicitis** | | | | | | | | | | | | |
| First dose/positive test (before vaccine) | 63 | 1.02 (0.78, 1.35) | 0 | 0 | - | - | 0 | - | - | 213 | 1.15 (0.99, 1.33) | 0 |
| Second dose/positive test (after vaccine) | 8 | 0.80 (0.39, 1.65) | 0 | 0 | - | - | 0 | - | - | <5 | - | - |
| Third dose | 0 | - | - | 0 | - | - | 0 | - | | | | |
| **Guillain-Barre Syndrome** | | | | | | | | | | | | |
| First dose/positive test (before vaccine) | 0 | - | - | 0 | - | - | 0 | - | - | 0 | - | - |

**Table 3 (continued) | Incidence rate ratios and estimated number of excess events per million for individual outcomes in the 1–42 days following vaccination or a SARS-CoV-2 positive test (before or after vaccination) compared to baseline period in children aged 5-11 years**

| | BNT162b2 vaccine | | | mRNA-1273 vaccine | | | ChAdOX1 vaccine | | | Positive SARS-CoV-2 test | | |
|---|---|---|---|---|---|---|---|---|---|---|---|---|
| | N | IRR (95% CI) | Excess events (95% CI) | N | IRR (95% CI) | Excess events (95% CI) | N | IRR (95% CI) | Excess events (95% CI) | N | IRR (95% CI) | Excess events (95% CI) |
| Second dose/positive test (after vaccine) | 0 | - | - | 0 | - | - | 0 | - | - | 0 | - | - |
| Third dose | 0 | - | - | 0 | - | - | 0 | - | - | | | |
| **Angioedema** | | | | | | | | | | | | |
| First dose/positive test (before vaccine) | < 5 | - | - | 0 | - | - | 0 | - | - | 12 | 1.01 (0.55, 1.87) | 0 |
| Second dose/positive test (after vaccine) | 0 | - | - | 0 | - | - | 0 | - | - | 0 | - | - |
| Third dose | 0 | - | - | 0 | - | - | 0 | - | - | | | |
| **Anaphylaxis** | | | | | | | | | | | | |
| First dose/positive test (before vaccine) | < 5 | - | - | 0 | - | - | 0 | - | - | 24 | 1.45 (0.91, 2.31) | 0 |
| Second dose/positive test (after vaccine) | 0 | - | - | 0 | - | - | 0 | - | - | 0 | - | - |
| Third dose | 0 | - | - | 0 | - | - | 0 | - | - | | | |

*N* Number of events, *IRR* Incidence rate ratio, *CI* Confidence interval.
Cells with * are models that did not converge.

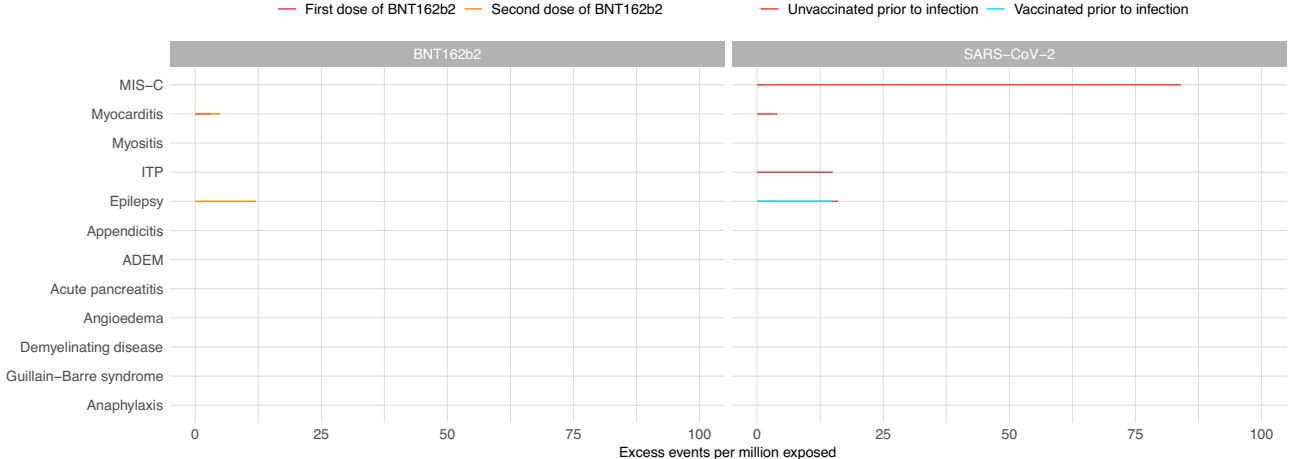

**Fig. 1 | Risk of serious outcomes following a first or second dose of COVID-19 vaccination with the BNT162b2 vaccine or SARS-CoV-2 infection (prior to vaccination) in adolescents aged 12–17 years.** Estimated number of excess events per million (95% CI) based on incidence rate ratios of each outcome in the 1–42 days following vaccination or SARS-CoV-2 positive test compared to baseline period are presented where there were at least five events during the exposure period and when number of excess events is greater than zero. Data available from 8th December 2020 and 7th August 2022. Table 4 contains the data presented in this figure. MIS-C Multisystem inflammatory syndrome; ITP Idiopathic or immune thrombocytopenic purpura, ADEM Acute disseminated encephalomyelitis.

## Matched cohort study

Our matched cohort analysis included 1,580,869 children aged 5–11 years and 1,535,341 adolescents aged 12–17 years. Characteristics of the cohort are detailed in Supplementary Table 10.

Incidence rates of vaccine safety outcomes in the 1–42 days following each vaccine dose and following SARS-CoV-2 infection in vaccinated and unvaccinated children are presented in Supplementary Table 11. Incidence rates for all outcomes were significantly higher following SARS-CoV-2 infection compared to COVID-19 vaccination.

We matched 160,262 children aged 5–11 years and 848,186 adolescents aged 12–17 years who had received at least one dose of COVID-19 vaccine to a child of the same age and sex who had not received any COVID-19 vaccine doses by the date of the vaccinated child's first vaccine dose (characteristics of matched cohort reported in Supplementary Table 12).

As in the self-controlled case series analysis, we identified an increased risk of hospitalisation with epilepsy in the 1–42 days following a second dose of COVID-19 vaccine with BNT162b2 in 12-17-year-olds (unadjusted IRR 1.77, 95%CI 1.05–2.99, adjusted IRR 3.88, 95% CI 1.27–11.86), but did not find significantly increased risks of appendicitis or myocarditis with BNT162b2 vaccination in adolescents (Supplementary Table 13).

We identified additional increased risks of anaphylaxis and appendicitis in 12-17-year-olds following a first dose of BNT162b2 (unadjusted IRR 3.71, 95%CI 1.23–11.14 and unadjusted IRR 1.37, 95%CI 1.05–1.80, respectively) and an increased risk of hospitalisation with epilepsy following a first dose with BNT162b2 in 5–11-year-olds, although the confidence interval was very wide reflecting the uncertainty of the estimate (unadjusted IRR 16.00, 95%CI 2.12–120.65) (Supplementary Table 13).

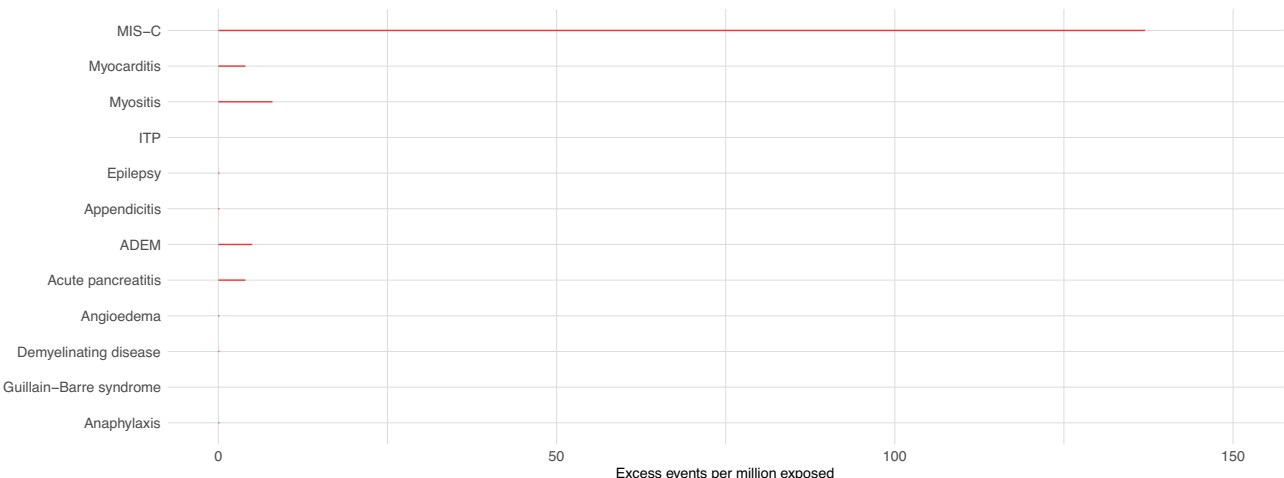

**Fig. 2 | Risk of serious outcomes following SARS-CoV-2 infection prior to vaccination in children aged 5–11 years.** Estimated number of excess events per million (95% CI) based on incidence rate ratios of each outcome in the 1–42 days following SARS-CoV-2 positive test compared to baseline period are presented where there were at least five events during the exposure period and when number of excess events is greater than zero. Data available from 8th December 2020 and 7th August 2022. Table 3 contains the data presented in this figure. MIS-C Multisystem inflammatory syndrome, ITP Idiopathic or immune thrombocytopenic purpura, ADEM Acute disseminated encephalomyelitis.

In general, the estimates from the matched cohort study were in agreement with the results from the self-controlled case series analysis in under-18s.

Unadjusted IRRs and IRRs adjusted for self-reported ethnicity (white, non-white, missing), quintile of deprivation (based on Townsend score) and presence of comorbidity (yes/no) for each outcome are reported in Supplementary Table 13.

## Discussion

In our study of approximately 5.1 million young people aged 5–17 years, we have examined the risks of vaccine safety outcomes following COVID-19 vaccination and SARS-CoV-2 infection, stratified by age group (5–11 years and 12–17 years) and additionally by sex (males and females). We report several key findings that are relevant to public health policy makers. Firstly, we found no strong evidence for increased risks of any of the 12 safety outcomes investigated following COVID-19 vaccination in children aged 5–11 years. Second, in adolescents aged 12–17 years, we observed an increased risk of hospital admission for myocarditis following a first or second dose of BNT162b2, and an increased risk of hospitalisation with epilepsy and demyelinating disease (in females only) following a second dose of BNT162b2 in our primary analysis. Third, children and adolescents aged 5–17 years who had not received a COVID-19 vaccine dose prior to SARS-CoV-2 infection had an increased risk of hospitalisation from seven of the pre-specified safety outcomes including MIS-C and myocarditis. Finally, the risks of safety outcomes from SARS-CoV-2 infection were largely absent in 5–17-year-olds who received at least one COVID-19 vaccine dose prior to infection.

In our cohort of > 2.8 million vaccinated adolescents aged 12–17 years, we estimated 3 (95%CI 0–5) and 5 (95%CI 3–6) excess cases of myocarditis per million exposed in the 1–42 days following a first and second dose of BNT162b2, respectively. We did not observe an increased risk of myocarditis following vaccination with mRNA-1273 in adolescent males or females as some previous studies have reported[23–32]. However, the mRNA-1273 vaccine was only given to a relatively small number of adolescents in England, primarily for the third booster dose, therefore the study was underpowered to detect statistically significant associations, except for very large effect sizes. As expected, we observed a substantially increased risk of myocarditis in the 1–42 days following almost all doses of both mRNA vaccines in young adults aged 18–24 years compared to the baseline period,

consistent with other studies reporting an increased risk of myocarditis in young adult males following a second mRNA-1273 vaccine dose[24–26]. Importantly, however, there were no deaths following a diagnosis of myocarditis in under-18s, and our findings are likely to reflect self-limiting disease.

We also observed a modest increased risk of hospitalisation with epilepsy in adolescents following a second dose of BNT162b2, with an additional 12 (95%CI 0–23) cases estimated per million exposed in 12–17-year-olds. However, a diagnosis of epilepsy is made over a period of time as it typically involves outpatient referral, a magnetic resonance imaging (MRI) scan and an electroencephalogram[47]. Therefore, this reported increased risk of epilepsy is highly unlikely to reflect new-onset epilepsy triggered by the vaccine. A limitation of the data was that we were only able to exclude prior hospital admissions with epilepsy (or any of the pre-specified outcomes) in the two years preceding the study start date. Therefore, the increase in admissions reported here is more likely to reflect seizures in children with an already underlying diagnosis of epilepsy or other chronic neurological condition but who hadn't been hospitalised in the previous two years, which we were unable to capture in our dataset, rather than new diagnoses.

Seizure exacerbation following COVID-19 vaccination has been reported in previous studies, but primarily in adults with epilepsy rather than children[40]. One survey of 224 children (median age 8 years) with epilepsy reported that 10% of those who had been living with epilepsy for more than two years or who had not been seizure-free in the year prior to vaccination had seizures in the month following the first and second dose of COVID-19 vaccine[48]. However, in children who were seizure-free for at least six months before they were vaccinated, the risk of seizures was decreased following vaccination[48]. Another survey of 278 people with Dravet Syndrome (with a median age 19 years) showed that there was increased seizure frequency following COVID-19 vaccination in 13% of the 120 participants who had received at least one vaccine dose[49]. Our findings suggest that the BNT162b2 vaccine is associated with a small increased risk of hospitalisation with epilepsy, but this risk is most likely restricted to children with pre-existing epilepsy and should be balanced against risks of hospitalisation through natural infection with SARS-CoV-2.

In the sex-stratified analysis, we identified a small increased risk of demyelinating disease in females following a second dose of BNT162b2, with an estimated 4 (95%CI 0–6) excess cases per million. To date there has been weak evidence linking mRNA vaccines with

**Table 4 | Incidence rate ratios and estimated number of excess events per million for individual outcomes in the 1–42 days following vaccination or a SARS-CoV-2 positive test (before or after vaccination) compared to baseline period in adolescents aged 12-17 years**

| | BNT162b2 vaccine | | | mRNA-1273 vaccine | | | ChAdOX1 vaccine | | | Positive SARS-CoV-2 test | | |
|---|---|---|---|---|---|---|---|---|---|---|---|---|
| | N | IRR (95% CI) | Excess events (95% CI) | N | IRR (95% CI) | Excess events (95% CI) | N | IRR (95% CI) | Excess events (95% CI) | N | IRR (95% CI) | Excess events (95% CI) |
| **Multisystem inflammatory syndrome** | | | | | | | | | | | | |
| First dose/positive test (before vaccine) | 21 | 0.76 (0.43, 1.37) | 0 | 0 | - | - | 0 | - | - | 98 | 12.38 (8.88, 17.28) | 84 (81, 86) |
| Second dose/positive test (after vaccine) | <5 | - | - | 0 | - | - | 0 | - | - | <5 | - | - |
| Third dose | <5 | - | - | 0 | - | - | 0 | - | - | | | |
| **Myocarditis** | | | | | | | | | | | | |
| First dose/positive test (before vaccine) | 19 | 1.92 (1.08, 3.43) | 3 (0, 5) | 0 | - | - | <5 | - | - | 7 | 3.17 (1.34, 7.50) | 4 (2, 6) |
| Second dose/positive test (after vaccine) | 17 | 2.96 (1.65, 5.32) | 5 (3, 6) | 0 | - | - | 0 | - | - | <5 | - | - |
| Third dose | <5 | - | - | <5 | - | - | 0 | - | - | | | |
| **Immune or idiopathic thrombocytopenia** | | | | | | | | | | | | |
| First dose/positive test (before vaccine) | 41 | 0.76 (0.55, 1.07) | 0 | 0 | - | - | <5 | - | - | 37 | 1.79 (1.26, 2.54) | 15 (7, 21) |
| Second dose/positive test (after vaccine) | 31 | 0.83 (0.57, 1.21) | 0 | 0 | - | - | <5 | - | - | 6 | 1.64 (0.72, 3.74) | 0 |
| Third dose | 10 | 0.72 (0.37, 1.37) | 0 | <5 | - | - | 0 | - | - | | | |
| **Epilepsy** | | | | | | | | | | | | |
| First dose/positive test (before vaccine) | 215 | 1.02 (0.88, 1.18) | 0 | 0 | - | - | 15 | 1.93 (1.10, 3.39) | 705 (129, 1033) | 77 | 1.29 (1.02, 1.64) | 16 (2, 28) |
| Second dose/positive test (after vaccine) | 190 | 1.17 (1.00, 1.37) | 12 (0, 23) | <5 | - | - | 9 | 1.19 (0.59, 2.41) | - | 22 | 1.58 (1.03, 2.42) | 15 (1, 24) |
| Third dose | 48 | 0.83 (0.62, 1.13) | 0 | 6 | 0.95 (0.38, 2.36) | 0 | 0 | - | - | | | |
| **Acute pancreatitis** | | | | | | | | | | | | |
| First dose/positive test (before vaccine) | 28 | 1.25 (0.82, 1.92) | 0 | 0 | - | - | <5 | - | - | 7 | 1.04 (0.47, 2.27) | 0 |
| Second dose/positive test (after vaccine) | 10 | 0.65 (0.34, 1.25) | 0 | 0 | - | - | 0 | - | - | <5 | - | - |
| Third dose | 5 | 0.87 (0.34, 2.20) | 0 | 0 | - | - | 0 | - | - | | | |
| **Myositis** | | | | | | | | | | | | |
| First dose/positive test (before vaccine) | 7 | 1.20 (0.53, 2.75) | 0 | 0 | - | - | 0 | - | - | <5 | - | - |
| Second dose/positive test (after vaccine) | <5 | - | - | 0 | - | - | <5 | - | - | 0 | - | - |
| Third dose | <5 | - | - | 0 | - | - | 0 | - | - | | | |
| **Acute disseminated encephalomyelitis** | | | | | | | | | | | | |
| First dose/positive test (before vaccine) | 5 | 0.61 (0.24, 1.56) | 0 | 0 | - | - | 0 | - | - | 6 | 1.29 (0.55, 3.02) | 0 |

**Table 4 (continued) | Incidence rate ratios and estimated number of excess events per million for individual outcomes in the 1–42 days following vaccination or a SARS-CoV-2 positive test (before or after vaccination) compared to baseline period in adolescents aged 12-17 years**

| | BNT162b2 vaccine | | | mRNA-1273 vaccine | | | ChAdOX1 vaccine | | | Positive SARS-CoV-2 test | | |
|---|---|---|---|---|---|---|---|---|---|---|---|---|
| | N | IRR (95% CI) | Excess events (95% CI) | N | IRR (95% CI) | Excess events (95% CI) | N | IRR (95% CI) | Excess events (95% CI) | N | IRR (95% CI) | Excess events (95% CI) |
| Second dose/positive test (after vaccine) | 8 | 1.46 (0.67, 3.18) | 0 | 0 | - | - | 0 | - | - | <5 | - | - |
| Third dose | 0 | - | - | 0 | - | - | 0 | - | - | | | |
| **Demyelinating disease** | | | | | | | | | | | | |
| First dose/positive test (before vaccine) | 7 | 0.85 (0.38, 1.89) | 0 | 0 | - | - | 0 | - | - | <5 | - | - |
| Second dose/positive test (after vaccine) | 9 | 1.55 (0.75, 3.21) | 0 | 0 | - | - | 0 | - | - | <5 | - | - |
| Third dose | <5 | - | - | 0 | - | - | 0 | - | - | | | |
| **Appendicitis** | | | | | | | | | | | | |
| First dose/positive test (before vaccine) | 461 | 0.90 (0.81, 1.00) | 0 | 0 | - | - | <5 | - | - | 209 | 1.07 (0.93, 1.24) | 0 |
| Second dose/positive test (after vaccine) | 330 | 0.89 (0.79, 1.01) | 0 | 0 | - | - | 6 | 4.64 (1.77, 12.17) | 512 (283, 599) | 26 | 0.80 (0.54, 1.19) | 0 |
| Third dose | 64 | 1.04 (0.80, 1.35) | 0 | 9 | 1.27 (0.64, 2.52) | 0 | 0 | - | - | | | |
| **Guillain-Barre Syndrome** | | | | | | | | | | | | |
| First dose/positive test (before vaccine) | <5 | - | - | 0 | - | - | 0 | - | - | 5 | - | - |
| Second dose/positive test (after vaccine) | <5 | - | - | 0 | - | - | 0 | - | - | <5 | - | - |
| Third dose | 0 | - | - | 0 | - | - | 0 | - | - | | | |
| **Angioedema** | | | | | | | | | | | | |
| First dose/positive test (before vaccine) | 25 | 1.40 (0.89, 2.19) | 0 | 0 | - | - | <5 | - | - | 11 | 1.46 (0.77, 2.76) | 0 |
| Second dose/positive test (after vaccine) | 12 | 0.91 (0.50, 1.68) | 0 | 0 | - | - | <5 | - | - | <5 | - | - |
| Third dose | <5 | - | - | <5 | - | - | 0 | - | - | | | |
| **Anaphylaxis** | | | | | | | | | | | | |
| First dose/positive test (before vaccine) | 44 | 1.33 (0.94, 1.87) | 0 | 0 | - | - | 0 | - | - | 19 | 1.50 (0.92, 2.42) | 0 |
| Second dose/positive test (after vaccine) | 21 | 0.85 (0.54, 1.34) | 0 | 0 | - | - | 0 | - | - | <5 | - | - |
| Third dose | 7 | 1.26 (0.57, 2.79) | 0 | <5 | - | - | 0 | - | - | | | |

N Number of events, IRR Incidence rate ratio, CI Confidence interval.

neuroimmunological disorders[50,51], and a similar self-controlled case series in adults only identified links between the ChAdOX1 vaccine and neuroimmunological conditions[6]. The coding description *demyelinating disease* can refer to a spectrum of monophasic and chronic neuroinflammatory disorders, however, the majority of cases following BNT162b2 were coded as optic neuritis, which is typically monophasic and associated with good visual recovery in most childhood cases[52]. From the neurological perspective, this risk should be balanced against the protection vaccination offers against rare and more severe neuroimmunological sequelae of COVID-19 infection such as ADEM.

This study identified two strong safety signals in adolescents associated with the ChAdOX1 vaccine: appendicitis and epilepsy. A substantially increased risk of appendicitis was observed in adolescents following a second dose of ChAdOX1, with an additional 512 (95% CI 283–599) cases expected per million. This estimate is based on a small sample size as the ChAdOX1 vaccine was not approved for use in under-40s in the UK from April 2021[3,53]. Additionally, the increased risk was not identified in the matched cohort study, suggesting the evidence from this study for a causal association between appendicitis and ChAdOX1 vaccination is weak. Appendicitis was highlighted as an outcome of interest for vaccine safety by the US Food and Drug Administration following a clinical trial of BNT162b2, which reported a higher number of appendicitis cases in the vaccine arm compared to the placebo arm[44]. However, subsequent evidence from observational studies and adverse event reporting databases is conflicting, limited to adults and primarily focusing on mRNA vaccines[11,54,55].

Following a first dose of ChAdOX1, we observed a substantially increased risk of hospitalisation with epilepsy, particularly in females, with 813 (95% CI 44–1164) excess cases estimated per million female adolescents vaccinated with a first dose of ChAdOX1, but as discussed above, these are not likely to reflect new diagnoses of epilepsy. The small sample size (only 0.3% of adolescents received ChAdOX1 for the first dose) and resulting wide confidence interval for this estimate should be noted. We also found that there was a higher proportion of adolescents with a hospital admission for epilepsy in the two years prior to the study start date who received ChAdOX1 for the first dose (2.7%), compared to BNT162b2 (0.2%) and mRNA-1273 (0.6%). While these individuals were excluded from the analysis, it is indicative that a higher proportion of adolescents who received a ChAdOX1 vaccine were included in a priority group, such as those with a chronic neurological disease including epilepsy[3], compared to those who received mRNA vaccines and that the majority of hospitalisations with epilepsy following ChAdOX1 were likely in adolescents with a pre-existing condition. We did not identify an increased risk of hospitalisation with epilepsy following vaccination with ChAdOX1 in the matched cohort study. Our findings, together with a recent study that found evidence for increased risk of cardiac death in young women following a first dose of non-mRNA vaccine[56], suggest that further work would need to be done to ensure the safety of ChAdOX1 in young people if it were to be used in future vaccination programmes.

In our cohort of > 580,000 children aged 5–11 years who were vaccinated against COVID-19, we found no evidence for increased risks of any of the pre-specified adverse events following any dose of BNT162b2, mRNA-1273 or ChAdOX1 vaccine[33–35]. Based on vaccine uptake rates reported by the UK Health Security Agency[2], this analysis includes most vaccinated children in this age group in England, however, the lack of safety signals identified in this analysis could partially reflect the relatively small sample size due to the low uptake rate of COVID-19 vaccines in 5–11-year-olds. In the matched cohort study including > 160,000 vaccinated 5–11-year-olds, we additionally found an increased risk of hospital admission with epilepsy following a first dose with BNT162b2 compared to unvaccinated children. However, given that epilepsy was not identified as a safety signal in this age group in the self-controlled case series analysis and the lengthy

diagnosis pathway for epilepsy as described above, this potential increased risk is most likely restricted to children with pre-existing epilepsy.

Following SARS-CoV-2 infection we observed increased risks of hospital admission from MIS-C, myocarditis, acute pancreatitis, myositis and ADEM in children aged 5–11 years who had not been vaccinated against COVID-19 prior to infection. Most notably, we estimated an additional 137 (95%CI 134–140) hospital admissions from MIS-C in the four to six weeks after the date of a positive test being reported per million children exposed. We also observed increased risks of MIS-C, myocarditis, ITP and hospitalisation with epilepsy in adolescents aged 12–17 years following SARS-CoV-2 infection in those who had not been vaccinated prior to infection. 84 (95%CI 81-86) additional cases of MIS-C per million exposed would be expected following SARS-CoV-2 infection in adolescents of this age group prior to vaccination.

However, in both children and adolescents, these increased risks of serious outcomes from SARS-CoV-2 infection were absent in those who received at least one dose of COVID-19 vaccine prior to infection. The exception was hospitalisation with epilepsy. Our analyses suggested that the risk of admission to hospital with epilepsy was increased following SARS-CoV-2 infection in adolescents aged 12–17 years who were not vaccinated against COVID-19 prior to infection, with an estimated additional 16 (95%CI 2–28) cases per million. A risk was still seen following infection in those who received at least one vaccine dose prior to infection, but at a slightly lower level, with an additional 15 (95%CI 1–24) hospital admissions with epilepsy per million. This study has shown that vaccination is associated with a significantly reduced risk of most SARS-CoV-2 complications in young people, particularly MIS-C, which can be fatal[15].

This study has several strengths. First, this was a population-based study of prospectively recorded data not subject to recall and selection biases linked to case reports. Second, the large sample size allowed us to investigate rare outcomes, particularly following vaccination with BNT162b2, which could not be assessed through clinical trials. Third, the self-controlled case series study design removes potential confounding from fixed characteristics, and the additional breakdown of our study period into weekly blocks accounted for temporal confounding. We also assessed the robustness of our results through several sensitivity analyses, a matched cohort analysis and a parallel analysis in 18–24-year-olds to ensure that the results from our study were consistent with the current evidence base in adults.

There were also some limitations to this study. First, despite the large sample size, this study may have been under-powered to investigate rare outcomes in 5–11-year-olds due to the relatively small proportion of vaccinated children in this age group. Second, we relied on hospital admission codes and death certification to define the outcomes so our design will not have captured milder events only occurring in the community or only reported in GP records. This could have resulted in a misclassification bias. Third, we only had access to cases of SARS-CoV-2 infection confirmed with a reverse transcription polymerase chain reaction (RT-PCR) test in our database, which were more likely to happen in the early stages of the pandemic or in a hospital setting, and less likely to be used in schools and community settings where the vast majority of routine testing took place. We were also unable to account for unascertained SARS-CoV-2 infections, therefore, some of the adverse events could be misclassified as being associated with vaccination rather than a SARS-CoV-2 infection that was not recorded and the analysis of adverse outcomes following SARS-CoV-2 infection may have been biased by incomplete COVID-19 testing in the English population. Fourth, we were unable to determine the effect of SARS-CoV-2 variant on the risk of adverse events, as detailed data on the viral variant of SARS-CoV-2 underlying recorded infections was also not available in our database. Fifth, although we adjusted for seasonal effects in the self-controlled case series models, we did not explicitly investigate the effect of the wave of the

pandemic during which the infection occurred on the risk of adverse events. For example, the incidence of MIS-C has been reported to be lower during the periods when Delta and Omicron variants were dominant, even before 12–15-year-olds started being vaccinated, compared to the period when the Alpha variant was dominant[57]. Given that > 98% of children in this study were vaccinated against COVID-19 during the periods that Delta and Omicron were dominant, we may expect a lower incidence of MIS-C following infection in vaccinated children, who were unlikely to have been infected with the Alpha variant, compared to unvaccinated children, who had a higher likelihood of being infected with the Alpha variant. Lastly, we were unable to assess differences in vaccine safety by level of deprivation, which should be prioritised as a future area of research.

In summary, we found no strong evidence for increased risks of 12 pre-specified vaccine safety outcomes following COVID-19 vaccination in children aged 5–11 years and no new significant safety concerns in 12–17-year-olds following vaccination with mRNA vaccines recommended for use in these age groups in the UK by the JCVI. Additionally, in unvaccinated children we found increased risks of hospitalisation from seven adverse outcomes including MIS-C and myocarditis following SARS-CoV-2 infection that were either not observed, or were reduced, following vaccination. Overall, our findings support a favourable safety profile of COVID-19 vaccination using mRNA vaccines in children and young people aged 5-17 years.

## Methods

### Ethics approval and consent to participate
The QResearch® ethics approval was provided by the East Midlands-Derby Research Ethics Committee [reference 18/EM/0400] and reviewed by the QResearch science committee [Project OX300]. Consent from participants was not required. The study was performed in accordance with the Declaration of Helsinki.

### Data sources
We used the NIMS database of COVID-19 vaccination. We linked individual-level vaccination data to national data for mortality (Office for National Statistics), hospital admissions (Hospital Episode Statistics), and SARS-CoV-2 infection assessed by RT-PCR (Second Generation Surveillance System) using the unique NHS number. Additional analyses to assess robustness of the results made use of the QResearch database, which includes anonymised health records from ~1800 family practices across England.

### Study design
We undertook a self-controlled case series design, originally developed to examine vaccine safety[36,37], to explore the association between BNT162b2, mRNA-1273 and ChAdOx1 vaccines and pre-specified outcomes. Separate analyses were conducted for each outcome. The analyses were conditional on each case, thus any fixed characteristic during the study period, such as sex and ethnicity, were inherently controlled for by design. Age was considered as a fixed variable because the study period was short.

### Study period and population
The cohort included all children aged 5–17 years who had received at least one dose of BNT162b2, mRNA-1273 or ChAdOx1 vaccine or who had a positive SARS-CoV-2 test between 8th December 2020 and 7th August 2022. In each self-controlled case series analysis, we only included children who were admitted to hospital or died from the outcome during the study period and excluded children with a hospitalisation for the same outcome in the two years prior to 8th December 2020 and those who received other COVID-19 vaccine types. We also undertook an analysis in young adults aged 18–24 years as a comparison.

### Outcome
We a priori selected severe outcomes resulting in hospital admission or death which are monitored by national medical regulatory authorities, clinical trials, post-marketing surveillance, emerging scientific literature and neurological, cardiology, paediatric and immunology/vaccine expertise available. Our pre-defined outcomes were previously reported COVID-19 infection- and COVID-19 vaccination-related adverse events with strong evidence in young people: myocarditis[21,22], MIS-C[38] and myositis[45] and those in adults: Guillain-Barre syndrome[43], demyelinating disease[6], ITP[39] and appendicitis[44] as well as adverse events reported following any vaccination during childhood and early young adulthood: epilepsy[40], acute pancreatitis[41], ADEM[42] and angioedema[46]. The selected outcomes are consistent with previous studies reporting COVID-19 vaccine safety in young people[58]. Anaphylaxis was also included as a positive control outcome because it can occur shortly after vaccination[46]. Outcomes were identified using relevant International Classification of Diseases codes (https://www.qresearch.org/data/qcode-group-library/). For each outcome, we used the first event recorded during the study period and did not incorporate readmissions for the same outcome into the analysis.

### Exposures
Our main vaccine exposures were a first, second and third dose of BNT162b2, mRNA-1273 or ChAdOx1. To account for heterologous vaccination, each vaccine type and dose was considered separately. SARS-CoV-2 infection exposure was defined as the first positive SARS-CoV-2 test (assessed by RT-PCR) within the study period and was included as a separate variable, allowing safety outcomes that occurred following vaccination and infection to be allocated to both exposures in the model and mutually adjusted. We did not include reinfections in the analysis. We distinguished between infection occurring before the first vaccine dose, or after. We defined the exposure risk period as 1–42 days after each exposure under the assumption that the adverse events under consideration were unlikely to be related to the exposures if they occurred after 42 days. In the case where two vaccine doses were given within 42 days of each other, the risk period of the earlier dose was truncated on the date prior to the day that the next vaccine dose was received. People with a recent hospital admission may delay vaccination, therefore a pre-risk period of 1–28 days before each exposure was excluded from the baseline period to account for this potential bias[36]. Since people may have SARS-CoV-2 testing on hospital admission, we allocated day zero to a risk period of its own[36]. Whilst these admissions may have been caused by SARS-CoV-2 infection, reverse causality involved in their detection could lead to overestimation of the effect of infection on the outcome. All remaining observation time was included in the baseline period.

### Statistical analysis
We analysed the data with the self-controlled case series method using a conditional Poisson regression model with an offset for the length of the exposure risk period. We fixed age as the age at date of first COVID-19 vaccination or date of first positive SARS-CoV-2 test recorded in study period for those who were unvaccinated. To allow for underlying seasonal effects, we split the study observation period into two-week periods and adjusted for these as a factor variable in the statistical models. We censored follow-up at the earliest of time of death, fourth dose or study end. We estimated the additional number of events per million persons exposed following vaccination or infection[59] and assessed significance at the 5% level.

In the primary analysis, we fitted the self-controlled case series model separately in children aged 5–11 years and adolescents aged 12–17 years and stratified by sex (male and female). We additionally fitted the model in 18–24-year-olds for comparison. As sensitivity analyses, we fitted the self-controlled case series model starting the

observation period at the day of first, second and third vaccine dose and without censoring for deaths due to the outcome, to ascertain the robustness of our results when the outcome increases the probability of death. We fitted the model excluding those who received a third dose, as they might be the most clinically vulnerable individuals. We also fitted the model including children who were vaccinated but did not have a record of a SARS-CoV-2 infection during the study period, to assess the effect of the vaccine alone, to account for the possibility that an adverse event could have occurred within 42 days of both a vaccine dose and a SARS-CoV-2 infection. Finally, we fitted the model including only SARS-CoV-2 positive patients who had not received any vaccine to assess the effect of SARS-CoV-2 exposure alone. We additionally tested the effect of ethnicity (white or non-white) on the risk of each outcome by including an interaction term between ethnicity and vaccine/infection exposures.

### Matched cohort analysis

We also conducted a *post hoc* matched cohort study using the QResearch database of primary care records, linked to hospital episode statistics, COVID-19 vaccination and SARS-CoV-2 infection data. The study population included all vaccinated children aged 5–17 years with GP records in the QResearch database, and matched unvaccinated children, irrespective of SARS-CoV-2 test status during the study period. We matched vaccinated children to children of the same age and sex who were unvaccinated at the time that the vaccinated child received their first dose (rounded to the nearest 7 days) at a ratio of 1:1. Unvaccinated children were sampled from the whole cohort and included children who were vaccinated later in the study period, and these children were censored on date of their first vaccination. The same matched pairs were included in the analysis of the second and third doses for those who received these doses, and where the matched unvaccinated child remained unvaccinated at the time of the second and third doses being received by the vaccinated child.

We estimated incidence rates and fitted conditional Poisson regression models to estimate IRRs of each outcome in the 1–42 days following a first, second or third dose of BNT162b2, mRNA-1273 or ChAdOX1 COVID-19 vaccine. We estimated unadjusted IRRs and IRRs adjusted for self-reported ethnicity (white, non-white, missing), quintile of deprivation (based on Townsend score) and presence of comorbidity (yes/no) that would result in inclusion in clinical risk group (diagnosis prior to vaccine being available).

We additionally conducted a matched cohort analysis in 18–24-year-olds for comparison.

Stata v17 was used for data analysis.

### Reporting summary

Further information on research design is available in the Nature Portfolio Reporting Summary linked to this article.

## Data availability

The data that support the findings of this study—NIMS Database of COVID-19, mortality (Office of National Statistics), hospital admissions (Hospital Episode Statistics), SARS-CoV-2 infection data (SGSS) and primary care (QResearch)—are not publicly available because they are based on deidentified national clinical records. Due to national and organizational data privacy regulations, individual-level data such as those used for this study cannot be shared openly. Access to the QResearch data is according to the information on the QResearch website (www.qresearch.org).

## Code availability

A sample of the code used for a similar study has been deposited in the public git repository of the research group, available at https://github.com/qresearchcode/COVID-19-vaccine-safety. This sample code can be used to run a self-controlled case series analysis in STATA.

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

## Acknowledgements

This research is funded by the NIHR School for Primary Care Research, Grant Reference Number 622. The views expressed are those of the author(s) and not necessarily those of the NIHR or the Department of Health and Social Care. We acknowledge the contribution of EMIS practices who contribute to QResearch and EMIS Health and the Universities of Nottingham and Oxford for expertise in establishing, developing or supporting the QResearch database. This project involves data derived from anonymised patient-level information collected by the NHS. The SARS-CoV-2 test data were originally collated, maintained and

quality assured by Public Health England (PHE) and transferred to NHS England during the study. Access to the data was therefore facilitated by NHS England. The Hospital Episode Statistics, Secondary Users Service (SUS-PLUS) datasets and civil registration data are used by permission from NHS England who retain the copyright in that data. NHS England and Public Health England bears no responsibility for the analysis or interpretation of the data. JHC is supported by an NIHR senior investigator award. NLM is supported by a British Heart Foundation Chair Award (CH/F/21/90010), Programme Grant (RG/20/10/34966) and Research Excellence Award (RE/18/5/34216). DPJH is supported by the Wellcome Trust (215621/Z/19/Z), Medical Research Foundation and UKDRI (principal funder UKRI Medical Research Council).

## Author contributions

M.P., E.C., J.H.C., C.A.C.C. led the study conceptualization, development of the research question and analysis plan and interpretation of the results. M.P. and J.H.C. obtained funding, obtained data approvals, undertook the data specification and curation. M.P. and E.C. designed the analysis, contributed to interpretation of the analysis, undertook data analysis and wrote the first draft of the paper. D.S., L.H., J.H., N.L.M., P.M., A.S., A.H., C.R. and D.P.J.H. contributed to the discussion on protocol development and provided critical feedback on drafts of the manuscript. All authors approved the protocol, contributed to the critical revision of the manuscript and approved the final version of the manuscript.

## Competing interests

J.H.C. reports grants from National Institute for Health Research (NIHR) Biomedical Research Centre, Oxford, John Fell Oxford University Press Research Fund, Cancer Research UK and Oxford Wellcome Institutional Strategic Support Fund and other research councils, during the conduct of the study outside the scope of this work. J.H.C. is founder and was shareholder until 9 Aug 2023 of ClinRisk Ltd, which produces open and closed source software to implement clinical risk algorithms (outside this work) into clinical computer systems. J.H.C. is an unpaid director of QResearch, a not-for-profit organisation which is a partnership between the University of Oxford and EMIS Health who supply the QResearch database used for this work and is a consultant for Endeavour Predict Ltd outside this work. A.S. serves on a number of UK and Scottish Government COVID-19 advisory groups and was a member of AstraZeneca's Thrombotic Thrombocytopenic Taskforce; all roles are unremunerated. A.H. is Deputy Chair of the Joint Committee on Vaccination and Immunisation. D.H. serves on the UK Government Commission on Human Medicines P.M. has received speaker and advisory board fees from Moderna and Astra Zeneca. All other authors declare no competing interests related to this paper.
