## [Peer Review File · Nature Communications]

Safety outcomes following COVID-19 vaccination and infection in 5.1 million childrenREVIEWER COMMENTS

Reviewer #1 (Remarks to the Author):

Review of "Risks and benefits of BNT162b2, mRNA-1273 and ChAdOX1 vaccines in 5.1 million children and young people"

OVERALL

This paper uses a national cohort of 5-17 year olds in England to understand risk/benefit of SARS-CoV-2 vaccination. It is well written and an important piece of work. It establishes that for 11 pre-defined adverse outcomes, the risks of the vaccine are outweighed by the benefits in averting some of the serious consequences of SARS-CoV-2 infection. An additional important finding, consistent with existing literature, is that there were no adverse events associated with vaccination in 5-11 year olds. Some of the observed differences by gender for different adverse outcomes were also new & interesting.

The findings are what would be expected given other studies on vaccine safety and efficacy in children and adolescents from around the world, but the novelty is in the self-controlled case-series design, the national cohort, the ability to break it down by vaccine dose and the selection of some less commonly studied adverse events (such as epilepsy and appendicitis)

However, there were areas where I thought the methods and the presentation of results were confusing, and some important limitations were not made explicit.

LARGER CONCERNS

Adverse outcomes

On page 3, lines 93-97, the authors state "our primary aim was to investigate and compare the risks of pre-specified outcomes following vaccination with BNT162b2, mRNA-1273 and ChAdOX1 in children. We also aimed to quantify the effects of COVID-19 vaccination against serious outcomes following SARS-CoV-2 infection in children, with a focus on MIS-C, as a guide to inform global public health policy considerations."

The abstract states the aim was "to compare risks of hospitalisation from adverse events after COVID-19 vaccination and infection"

The 11 adverse outcomes chosen were hospitalisation or death due to one of: myocarditis, MIS-C, immune thrombocytopenia (ITP), hospitalisation with epilepsy, acute pancreatitis, acute disseminated encephalomyelitis (ADEM), Guillain-Barre syndrome, appendicitis, demyelinating disease, myositis, angioedema and anaphylaxis.

What is missing from this list is hospitalisation directly due to Covid-19. Hospitalisation due to Covid-19 is one of the main adverse consequences of SARS-CoV-2 infection that vaccination is intended to prevent, and it seems an odd and large omission from the list considered. It seems that the authors sourced their list from known or suspected adverse consequences of SARS-CoV-2 vaccination (which obviously does not include covid hospitalisation) as described in their methods, but nonetheless hospitalisations (or at the very least, ICU admissions) due to Covid-19 should have been included. Without them, the title and abstract are potentially misleading since I think most people would read 'risk/benefit' and 'hospitalisations' and assume that hospitalisations directly due to Covid-19 would be included. Certainly, they are an important adverse outcome (see for instance, Wilde et al, Hospital admissions linked to SARS-CoV-2 infection in children: a cohort study of 3.2 million first ascertained infections in England, *BMJ*, 382:e073639, 2023").

I am not sure it is feasible for the authors to include Covid-19 hospitalisations at this stage, particularly as this would not now be a pre-defined outcome, but I think the title, abstract, and discussion need to make their exclusion much more transparent and be explicit (in more than one place) that the adverse outcomes considered were driven by consideration of those associated with vaccination, rather than SARS-CoV-2 infection. In fact, the introduction could be tweaked a bit to be explicit that this paper is designed to address vaccine safety concerns and not overall risk-

benefit of the vaccines (given no consideration of Covid-19 hospitalisation, long covid etc) and so only potential vaccine-associated adverse outcomes were considered.

In particular this sentence in the discussion (which is also echoed in the abstract) needs clarifying: "Altogether, our findings support a favourable risk-benefit profile of COVID-19 vaccination in under-18s using mRNA vaccines". Yes – these results certainly support a favourable risk-benefit but the benefit is considerably higher than reported here given that two of the main, potentially vaccine-preventable, adverse covid outcomes were not considered (Covid-19 hospitalisation and long covid).

Inclusion criteria

On page 11, lines 381-382, the authors state "We included all children aged 5-17 years who had received at least one dose of BNT162b2, mRNA-1273 or ChAdOx1 vaccine or had a positive SARS-CoV-2 test and were admitted to hospital or died from at least one of the outcomes between 8th December 2020 and 7th August 2022"

I read this as meaning all vaccinated children OR "children hospitalised or dead from one of the adverse outcomes after SARS-CoV-2 infection". However, the tables and supplementary material makes clear that all children with a positive SARS-CoV-2 PCR test were included (regardless of whether they were hospitalised). I'm not entirely sure how to read that sentence in the methods then, but it definitely has to be rewritten to make it clearer!

SMALLER QUERIES/CONCERNS:

Unascertained infections

The authors do not really discuss the important limitation of unascertained infections in their baseline set. These are clearly there, since the number of MIS-C cases the baseline cohort for SARS-CoV-2 infection (first column of supplementary table 4) seems too high otherwise plus of course we know that many children were not tested when symptomatic or had pauci- or asymptomatic infections. A previous undetected infection (more common in the first year of the infection when testing of children, particularly younger children, was much less widespread) could also act to reduce incidence of the adverse outcomes on subsequent infection and so make the effect of vaccination look weaker. There is also potential bias in which populations these unascertained infections occurred – it is plausible for instance that parents of vaccinated children are also more likely to test them for SARS-CoV-2 infection (whether asymptotically or symptomatically).

More detail in the methods

It would be good to get a bit more clarity on some of the methods. What did the team do with readmissions within 42 days of an exposure? Once a child had experienced an adverse outcome from an exposure (e.g. dose 1) were they removed from consideration for a second adverse outcome (e.g. from dose 2)? Were ages fixed at study start or first exposure (whether infection or vaccination)? Why were only PCR tests considered and not lateral flow tests? The latter would be particularly important for identifying infections in secondary school children for the year that there was regular testing within schools. (this should also be discussed in the limitations). What happened if there was a new recorded infection within 42 days of a vaccine dose? Was the adverse outcome assigned to infection or to vaccination? What if it was a second infection?

It would also be helpful to have a bit more information about the data sources used and linkage method. For instance, I assume mortality was from ONS and hospital data from HES inpatient data, but it should be stated explicitly. I also assume linkage was on NHS number, but again this should be stated.

Timing of vaccination and SARS-CoV-2 variant

While the authors looked at adverse events associated with infection disaggregated by whether infection was pre or post vaccination, there was no discussion on whether time since vaccination was considered (important given that vaccine protection does wane over time, e.g. Lin et al. Effects of Vaccination and Previous Infection on Omicron Infections in Children. New England Journal of Medicine 2022, doi:10.1056/NEJMc2209371.) And while adverse events after different doses of vaccine were explored, there was no discussion of strength of vaccine protection by dose

(eg. Vaccine protection will be stronger after 2 doses).

There was also no discussion of the potential impact of different SARS-CoV-2 variants on adverse outcomes and vaccine efficacy, which I think needs to be addressed, at least in the limitations.

Ethnicity and deprivation

We know that there have been differences in exposure and severity of illness by ethnicity and deprivation. The recent paper by Wilde et al (cited above) showed disproportionate severe illness (and MIS-C) in children of non white backgrounds and from more deprived areas. Why did the authors not consider disaggregating by these characteristics? I feel it should at least be mentioned in the discussion (since it might guide future vaccine policy on priority populations for instance).

Presentation of the results

Overall, as shown in Table 2, over 99% of children received the BNT162b2 vaccine as their first and second dose. To me then, the main and most reliable results from this paper pertain to BNT162b2. It would make the paper easier to follow and the tables MUCH easier to interpret in the main paper if results were only shown for the BNT162b2 vaccine in Table 3. Full results for the other two vaccines can be given in the supplementary and discussed briefly in the main paper, but the focus should be on BNT162b2, since that is where the overwhelming amount of data is.

Staying on Table 3, I think it would also be more transparent if all adverse outcomes were given for each subsection and "0" entered where there were no observed events. At the moment it is difficult to compare between age groups and conditions because each section has different outcomes. This would include a set of "0"s for SARS-CoV-2 vaccination in 5-11s.

Figures 1 and 2: I think a little more explanation in the caption would be helpful (e.g. be explicit on comparison to baseline). Figure 1 seems to have 3 colour dots (blue, green, red) but the legend only has 2 colours (blue and red). Be clear in the caption that absence of a dot is absence of events (I assume > e.g no unvax dot for epilepsy in women in 5-11s).

Figure 2 is lovely, but it would help to have gridlines, it's very hard to match the data on the right hand panel to the respective adverse outcome. Also I am not sure about the title which does not match the caption – it suggests it's just for infection but I think this figure shows after infection AND after each of the 3 possible doses (regardless of infection).

Supplementary Table 1 is a bit confusing. If this is the *Excluded* cohort surely it's people who did not receive a vaccine dose AND had no recorded SARS-CoV-2 infection? The title and the columns suggest it's an "or". This does not seem to match the stated study cohort (EITHER SARS-CoV-2 vaccine OR SARS-CoV-2 confirmed infection).

MINOR COMMENTS

- In the introduction (lines 55-56), the authors discuss vaccine coverage of 12-15 year olds but in the results use 12-17 year olds as the age cohort. This is fine, but then I think the introduction should discuss coverage in 12-17 year olds also (acknowledging that the vaccine offer and timing varied between 12-15 and 16-17 year olds – 16/17 year olds had access about 6 weeks earlier than 12-15 year olds).

- The authors say that there have been few studies on the safety of SARS-CoV-2 vaccines in 5-11 year olds at a large scale, but cite the systematic review by Watanabe et al. Assessment of Efficacy and Safety of MRNA COVID-19 Vaccines in Children Aged 5 to 11 Years: A Systematic Review and Meta-Analysis. JAMA Pediatrics 2023, doi:10.1001/jamapediatrics.2022.6243 (ref 33) which includes data on over 10 million vaccinated children. I think it's fair to say that the safety of vaccination in 5-11 yr olds is pretty well established.

- Results and discussion – while the data are available in the table, it would be good to compare risks of adverse outcomes explicitly between SARS-CoV-2 infection and vaccination, especially for myocarditis (given that this comparison is made in the paper for epilepsy).

- Lines 248-249: I found this sentence quite hard to parse and it wasn't immediately obvious to me that the authors were saying that rates of myocarditis were substantially higher in 18-24 year olds. I think because I read "in the comparison groups" as "vs the comparison group" but perhaps be explicit and say rates were higher in 18-24 than younger age groups.

- Final sentence: "This study highlights that the decision to vaccinate children and young people with COVID-19 vaccines should be carefully balanced with the risk of infection and complications following SARS-CoV-2 infection, as well as the broader impact of the COVID-19 pandemic on the well-being of young people." I'm not sure I think this sentence follows from this paper! All decisions to introduce a vaccine (whether for adults or children) require a careful risk-benefit analysis. And countries did carry those out. All the evidence since vaccination of children started is that adverse vaccine events are low and the risk of SARS-CoV-2 infection outweighs that of vaccination. This study adds to that evidence. There is no data in the paper on the wider impact of the pandemic (and are the authors thinking of things like long covid? Or home school? Or something else?) So I'm not sure what point the authors are making here?

- Table 3 has some cell entries saying " ≥ 10 " (e.g. ADEM) – why? There is nowhere in the methods that provides a reason for that? should it be a " $<$ " sign?

Reviewer #2 (Remarks to the Author):

I would like to thank this opportunity to review the draft entitled "Risks and benefits of BNT162b2, mRNA-1273 and ChAdOX1 vaccines in 5.1 million children and young people". The authors examined the association between COVID-19 vaccination and serious adverse outcomes in the UK nationwide linked data, using the self-controlled case series (SCCS) method. The draft is clearly written, whereas I have concerns in the current study.

Major point:

My strong suggestion is that the authors should conduct not only the SCCS analysis but also a traditional cohort analysis to report absolute incidence rates of the outcomes in children with and without vaccination, as well as their crude and adjusted incidence rate ratios.

I agree that the SCCS is generally more suitable for examining a causal relationship between an exposure such as vaccination and outcomes than the traditional cohort design. However, I have strong concerns whether the results of SCCS alone are used for discussing the risk-benefit balance of vaccination in the underlying population.

The SCCS uses only the data of patients with studied outcomes. The estimands of cohort studies and SCCS seem to be different; the former refers to the effect of the treatment in a population compared to a control group, and the latter to the within-subject effect (S. Greenland. A unified approach to the analysis of case-distribution (case-only) studies *Stat Med*, 18 (1) (1999), pp. 1-15). Although the authors estimated the excess events per million for each outcome based on the SCCS results, I am not certain whether this is equal to the (adjusted) excess risk comparing vaccinated and unvaccinated children in the underlying population, if the estimands are different between the SCCS and cohort analyses. While I read the reference 58 the authors cited (Wilson, K. & Hawken, S. Drug safety studies and measures of effect using the self controlled case series design. *Pharmacoepidemiol Drug Saf* 22, 108-110 (2013)), I was not sure whether this important point was mentioned and accounted for in their calculation of absolute risk in the cited paper. Also, the SCCS is not perfect: the SCCS has its own limitations and requires strong assumptions which may not be true. In particular, I wonder if children with the outcome of interest have the same probability of receiving vaccination, even after 28 days (as a "pre-risk" period) which the authors excluded from the analysis. i.e. it is possible that children with the outcome of interest, which may be "serious", are far unlikely to receive the COVID-19 vaccination later. A cohort study is unlikely to suffer from this bias, and therefore, anyway, the results of cohort study should be shown as part of the "triangulation" approach in epidemiology.

Minor point:

Lines 158-160 ("mRNA-1273 vaccine We found no evidence for significantly increased risks for any of the pre-specified outcomes in the 1-42 days following a first, second or third dose of mRNA-1273 vaccine in 12-17-year-olds.") could mislead some readers (who are not familiar with epidemiology) as if mRNA-1273 is safer than BNT162b2 vaccine. As the authors mention elsewhere in the draft, the detection of "statistical significance" largely depends on the number of children receiving the vaccine. The authors should be careful about this point when summarizing the results, and should not mislead some readers.

Reviewer #3 (Remarks to the Author):

General comments

In this article, the authors assessed a set of risks associated either to COVID-19 vaccination or SARS-CoV-2 infection in children and adolescents in the UK. Using a self-controlled design, they find that increased risks after infection - notably of myocarditis and MIS-C - outweigh the risks after vaccination. The question of the balance of risk and benefit of Covid vaccination in children and adolescents is very important and more debatable than that in adults.

I see two major concerns about the study findings.

1. Imperfect detection of SARS-CoV-2 infection

- Whereas vaccine exposure ought to be precisely measured, infection events are typically under reported, in that only a fraction of infections results in recorded confirmation tests. And this is likely differential with greater chance to confirm an infection in those with a severe and hospitalised outcome than with mild or asymptomatic forms. This would overestimate the risk after infection and the relative benefit of vaccination.
- The authors should better address this limitation and particularly discuss the relevance of the "million exposed" denominator of infected used in absolute risk calculation that provides support to their principal conclusion.

2. Time period of infections

- There is a major difference in symptoms occurrence between variants and the authors may have assessed serious outcomes post infection that are specific to variants of the first part of the study period.
- The effect of infection either before or after vaccination is confounded by the difference in pathogenicity between variants. See for example <https://doi.org/10.1093/cid/ciac553> where post infection risk of MIS-C is found lower with delta and omicron variants, even pre vaccination.
- This ought to be addressed to interpret the results.

A few minor points that could be considered to improve the paper:

- line 210: It may be helpful to remind the readers that the positive control outcome is anaphylaxis. And I cannot confirm from the tables that "the estimates agreed with the main results".

- line 237: I don't get this last part of the sentence. The sample of vaccinated children included in this study is either insufficient to observe any potential events (the authors do not seem to state that) or sufficient - which indicates no safety issue in the entire population. The authors should clarify the role of the uptake rate.

- line 273-275: This sentence is unclear and needs rewriting

- line 340: I would tone down this statement on "sufficient power". It seems surprising not to retrieve the association between mRNA-1273 vaccine and myocarditis in adolescents that was found in many settings.

- line 372: The authors should better explain the nature and use of the QResearch database.
- line 387: The authors should fully explain the selection process for the studied outcomes.
- line 399: How was handled the overlap of risk periods between dose 1 and 2 within 42 days in the SCCS method ?
- line 411-426: Did the SCCS model account for the violation of the assumption that the occurrence of an event does not influence subsequent exposure ? I would imagine that children developing one of the outcome after infection or previous vaccine dose would not receive a subsequent dose as planned.
- Table 5 has no column headings.

Safety outcomes following vaccination with BNT162b2, mRNA-1273 and ChAdOX1 COVID-19 vaccines and SARS-CoV-2 infection in 5.1 million children

Emma Copland, Martina Patone, Defne Saatci, Lahiru Handunnetthi, Jennifer Hirst, David P J Hunt, Nicholas L Mills, Paul Moss, Aziz Sheikh, Carol AC Coupland, Anthony Harnden, Chris Robertson, Julia Hippisley-Cox

Response to reviewer comments

Reviewer #1 (Remarks to the Author):

Review of “Risks and benefits of BNT162b2, mRNA-1273 and ChAdOX1 vaccines in 5.1 million children and young people”

OVERALL

This paper uses a national cohort of 5-17 year olds in England to understand risk/benefit of SARS-CoV-2 vaccination. It is well written and an important piece of work. It establishes that for 11 pre-defined adverse outcomes, the risks of the vaccine are outweighed by the benefits in averting some of the serious consequences of SARS-CoV-2 infection. An additional important finding, consistent with existing literature, is that there were no adverse events associated with vaccination in 5-11 year olds. Some of the observed differences by gender for different adverse outcomes were also new & interesting.

The findings are what would be expected given other studies on vaccine safety and efficacy in children and adolescents from around the world, but the novelty is in the self-controlled case-series design, the national cohort, the ability to break it down by vaccine dose and the selection of some less commonly studied adverse events (such as epilepsy and appendicitis)

However, there were areas where I thought the methods and the presentation of results were confusing, and some important limitations were not made explicit.

LARGER CONCERNS

Adverse outcomes

On page 3, lines 93-97, the authors state “our primary aim was to investigate and compare the risks of pre-specified outcomes following vaccination with BNT162b2, mRNA-1273 and ChAdOX1 in children. We also aimed to quantify the effects of COVID-19 vaccination against serious outcomes following SARS-CoV-2 infection in children, with a focus on MIS-C, as a guide to inform global public health policy considerations.”

The abstract states the aim was “to compare risks of hospitalisation from adverse events after COVID-19 vaccination and infection”

The 11 adverse outcomes chosen were hospitalisation or death due to one of:

myocarditis, MIS-C, immune thrombocytopenia (ITP), hospitalisation with epilepsy, acute pancreatitis, acute disseminated encephalomyelitis (ADEM), Guillain-Barre syndrome, appendicitis, demyelinating disease, myositis, angioedema and anaphylaxis.

What is missing from this list is hospitalisation directly due to Covid-19. Hospitalisation due to Covid-19 is one of the main adverse consequences of SARS-CoV-2 infection that vaccination is intended to prevent, and it seems an odd and large omission from the list considered. It seems that the authors sourced their list from known or suspected adverse consequences of SARS-CoV-2 vaccination (which obviously does not include covid hospitalisation) as described in their methods, but nonetheless hospitalisations (or at the very least, ICU admissions) due to Covid-19 should have been included. Without them, the title and abstract are potentially misleading since I think most people would read 'risk/benefit' and 'hospitalisations' and assume that hospitalisations directly due to Covid-19 would be included. Certainly, they are an important adverse outcome (see for instance, Wilde et al, Hospital admissions linked to SARS-CoV-2 infection in children: a cohort study of 3.2 million first ascertained infections in England, *BMJ*, 382:e073639, 2023”).

I am not sure it is feasible for the authors to include Covid-19 hospitalisations at this stage, particularly as this would not now be a pre-defined outcome, but I think the title, abstract, and discussion need to make their exclusion much more transparent and be explicit (in more than one place) that the adverse outcomes considered were driven by consideration of those associated with vaccination, rather than SARS-CoV-2 infection. In fact, the introduction could be tweaked a bit to be explicit that this paper is designed to address vaccine safety concerns and not overall risk-benefit of the vaccines (given no consideration of Covid-19 hospitalisation, long covid etc) and so only potential vaccine-associated adverse outcomes were considered.

In particular this sentence in the discussion (which is also echoed in the abstract) needs clarifying: “Altogether, our findings support a favourable risk-benefit profile of COVID-19 vaccination in under-18s using mRNA vaccines”. Yes – these results certainly support a favourable risk-benefit but the benefit is considerably higher than reported here given that two of the main, potentially vaccine-preventable, adverse covid outcomes were not considered (Covid-19 hospitalisation and long covid).

RESPONSE: Thank you for this comment. As the Reviewer states, the aim of our paper is to address COVID-19 vaccine safety concerns, and we have revised the wording of the title, aims, results and interpretation to reflect this. The efficacy of COVID-19 vaccines to protect against COVID-19 hospitalisation in children has been demonstrated in clinical trials and real-world effectiveness studies, therefore we prioritised this analysis of vaccine safety, as safety concerns have been identified as a barrier to vaccine uptake in children. As the Reviewer also suggests, we are unable to include COVID-19 hospitalisation at this stage of the project as it was not a pre-specified outcome.

The main changes we have made to reflect the focus on vaccine safety concerns are detailed below:

- The title now reads: *“Safety outcomes following vaccination with BNT162b2, mRNA-1273 and ChAdOX1 COVID-19 vaccines and SARS-CoV-2 infection in 5.1 million children”*
- The aims now state: *“Therefore, our primary aim was to investigate and compare the risks of pre-specified vaccine safety outcomes following vaccination with BNT162b2, mRNA-1273 and ChAdOX1 in children. We also aimed to compare the risks of these safety outcomes following SARS-CoV-2 infection in vaccinated and unvaccinated children, with a focus on MIS-C, as a guide to inform global public health policy considerations.”* [lines 101-105]
- The headings of the results section now read:

- *“Risk of pre-specified safety outcomes following COVID-19 vaccination”* [line 148]
- *“Risk of pre-specified safety outcomes following COVID-19 infection”* [line 206]
- The first sentence of the Discussion now reads: *“In our study of approximately 5.1 million children and young people aged 5-17 years, we have examined the risks of vaccine safety outcomes following COVID-19 vaccination and SARS-CoV-2 infection, stratified by age group (5-11 years and 12-17 years) and sex (males and females).”* [lines 290-292]
- The Interpretation now reads: *“In summary, we found no strong evidence for increased risks of 11 pre-specified vaccine safety outcomes following COVID-19 vaccination in children aged 5-11 years and no new significant safety concerns in 12-17-year-olds following vaccination with mRNA vaccines recommended for use in these age groups in the UK by the JCVI. Additionally, in unvaccinated children we found increased risks of hospitalisation from seven adverse outcomes including MIS-C and myocarditis following SARS-CoV-2 infection that were either not observed, or were reduced, following vaccination. Overall, our findings support a favourable safety profile of COVID-19 vaccination using mRNA vaccines in children and young people aged 5-17 years.”* [lines 449-456]

Inclusion criteria

On page 11, lines 381-382, the authors state “We included all children aged 5-17 years who had received at least one dose of BNT162b2, mRNA-1273 or ChAdOx1 vaccine or had a positive SARS-CoV-2 test and were admitted to hospital or died from at least one of the outcomes between 8th December 2020 and 7th August 2022”

I read this as meaning all vaccinated children OR “children hospitalised or dead from one of the adverse outcomes after SARS-CoV-2 infection”. However, the tables and supplementary material makes clear that all children with a positive SARS-CoV-2 PCR test were included (regardless of whether they were hospitalised). I’m not entirely sure how to read that sentence in the methods then, but it definitely has to be rewritten to make it clearer!

RESPONSE: We have clarified that the inclusion criteria apply to each individual self-controlled case series analysis, which only included cases, rather than the whole cohort. The Methods section now reads:

“The cohort included all children aged 5-17 years who had received at least one dose of BNT162b2, mRNA-1273 or ChAdOx1 vaccine or who had a positive SARS-CoV-2 test between 8th December 2020 and 7th August 2022. In each self-controlled case series analysis, we only included children who were admitted to hospital or died from the outcome during the study period and excluded children with a hospitalization for the same outcome in the two years prior to 8th December 2020 and those who received other COVID-19 vaccine types.” [lines 475-481]

SMALLER QUERIES/CONCERNS:

Unascertained infections

The authors do not really discuss the important limitation of unascertained infections in their baseline set. These are clearly there, since the number of MIS-C cases the baseline cohort for SARS-CoV-2 infection (first column of supplementary table 4) seems too high otherwise plus of course we know that many children were not tested when symptomatic or had pauci- or asymptomatic infections. A previous undetected infection (more common in the first year of the infection when testing of children, particularly younger children, was much less widespread) could also act to reduce incidence of the adverse outcomes on subsequent infection and so make the effect of vaccination

look weaker. There is also potential bias in which populations these unascertained infections occurred – it is plausible for instance that parents of vaccinated children are also more likely to test them for SARS-CoV-2 infection (whether asymptotically or symptomatically).

RESPONSE: Thank you for this comment, as we used routinely collected data, unascertained SARS-CoV-2 infections were indeed a limitation of this analysis. The effect of these unascertained infections could be an apparent increased risk of safety outcomes following vaccination, where outcomes have been misclassified as being associated with vaccination rather than a SARS-CoV-2 infection that was not recorded. This would be particularly relevant to outcomes that are known complications of COVID-19 infection, such as MIS-C, as pointed out by the reviewer. We have included this limitation in the Discussion section which now states:

“We were also unable to account for unascertained SARS-CoV-2 infections, therefore, some of the adverse events could be misclassified as being associated with vaccination rather than a SARS-CoV-2 infection that was not recorded and the analysis of adverse outcomes following SARS-CoV-2 infection may have been biased by incomplete COVID-19 testing in the English population.” [lines 431-435]

More detail in the methods

It would be good to get a bit more clarity on some of the methods. What did the team do with readmissions within 42 days of an exposure? Once a child had experience an adverse outcomes from an exposure (e.g. dose 1) were they removed from consideration for a second adverse outcomes (e.g. from dose 2)? Were ages fixed at study start or first exposure (whether infection or vaccination)? Why were only PCR tests considered and not lateral flow tests? The latter would be particularly important for identifying infections in secondary school children for the year that there was regular testing within schools. (this should also be discussed in the limitations). What happened if there was a new recorded infection within 42 days of a vaccine dose? Was the adverse outcome assigned to infection or to vaccination? What if it was a second infection?

RESPONSE: We have included this additional detail in the Methods and Discussion to clarify the study design and limitations.

- The Outcome section now states: *“For each outcome, we used the first event recorded during the study period and did not incorporate readmissions for the same outcome into the analysis.”* [lines 495-496]
- The Exposures section in Methods now reads:
 - o *“SARS-CoV-2 infection exposure was defined as the first positive SARS-CoV-2 test (assessed by RT-PCR) within the study period and was included as a separate variable, allowing safety outcomes that occurred following vaccination and infection to be allocated to both exposures in the model and mutually adjusted. We did not include reinfections in the analysis.”* [lines 500-503]
- The Statistical analysis section now reads:
 - o *“We fixed age as the age at date of first COVID-19 vaccination or date of first positive SARS-CoV-2 test recorded in study period for those who were unvaccinated.”* [lines 518-520]
 - o *“We also fitted the model including children who were vaccinated but did not have a record of a SARS-CoV-2 infection during the study period, to assess the effect of the vaccine alone, to account for the possibility that an adverse event could have occurred within 42 days of both a vaccine dose and a SARS-CoV-2 infection.”* [lines 532-535]

- The limitation section in the Discussion now reads: *“Third, we only had access to cases of SARS-CoV-2 infection confirmed with a reverse transcription polymerase chain reaction (RT-PCR) test in our database, which were more likely to happen in the early stages of the pandemic or in a hospital setting, and less likely to be used in schools and community settings where the vast majority of routine testing took place.”* [lines 427-431]

It would also be helpful to have a bit more information about the data sources used and linkage method. For instance, I assume mortality was from ONS and hospital data from HES inpatient data, but it should be stated explicitly. I also assume linkage was on NHS number, but again this should be stated.

RESPONSE: We have included this additional detail in the Data Sources section of the Methods which now states:

“We used the NIMS database of COVID-19 vaccination. We linked individual-level vaccination data to national data for mortality (Office for National Statistics), hospital admissions (Hospital Episode Statistics), and SARS-CoV-2 infection assessed by RT-PCR (Second Generation Surveillance System) using the unique NHS number.” [lines 461-464]

Timing of vaccination and SARS-CoV-2 variant

While the authors looked at adverse events associated with infection disaggregated by whether infection was pre or post vaccination, there was no discussion on whether time since vaccination was considered (important given that vaccine protection does wane over time, e.g. Lin et al. Effects of Vaccination and Previous Infection on Omicron Infections in Children. New England Journal of Medicine 2022, doi:10.1056/NEJMc2209371.) And while adverse events after different doses of vaccine were explored, there was no discussion of strength of vaccine protection by dose (eg. Vaccine protection will be stronger after 2 doses).

There was also no discussion of the potential impact of different SARS-CoV-2 variants on adverse outcomes and vaccine efficacy, which I think needs to be addressed, at least in the limitations.

RESPONSE: Thank you for this comment. As discussed in the response to the Reviewer’s first comment, evaluation of vaccine efficacy and waning was not an aim of this study as our focus was on vaccine safety concerns. We have added the limitation about lack of information on SARS-CoV-2 variants to the discussion, which now states:

“Fourth, we were unable to determine the effect of SARS-CoV-2 variant on the risk of adverse events, as detailed data on the viral variant of SARS-CoV-2 underlying recorded infections was also not available in our database.” [lines 435-437]

Ethnicity and deprivation

We know that there have been differences in exposure and severity of illness by ethnicity and deprivation. The recent paper by Wilde et al (cited above) showed disproportionate severe illness (and MIS-C) in children of non white backgrounds and from more deprived areas. Why did the authors not consider disaggregating by these characteristics? I feel it should at least be mentioned in the discussion (since it might guide future vaccine policy on priority populations for instance).

RESPONSE: Thank you for this suggestion. Unfortunately, data on deprivation level were not available for this analysis. However, we were able to evaluate whether the risks of vaccine safety

events varied by ethnicity, by including an interaction term between ethnicity and vaccine/COVID-19 exposure in a *post hoc* analysis. Ethnicity was categorised as white, non-white or missing and we reported the IRR of each outcome in people with non-white ethnicity relative to the IRR in people with white ethnicity following vaccination or infection. The results are presented in Supplementary Table 6 and detailed in the Results section:

“In a post hoc analysis investigating differences in risk between children of different ethnic backgrounds, we found that the risk of anaphylaxis following a second dose of BNT162b2 in adolescents with non-white ethnicity was higher relative to the risk in adolescents with white ethnicity (relative IRR 2.55, 95%CI 1.00-6.46) (Supplementary Table 6b). However, when the analysis was restricted to the subgroup of adolescents from non-white ethnic backgrounds, the risk of anaphylaxis following a second dose of BNT162b2 was not significantly increased compared to the baseline period (IRR 1.69, 95%CI 0.80, 3.54) (data not shown). We did not identify any differences in vaccine safety between white and non-white ethnicity for any of the other pre-specified outcomes.” [lines 179-187]

The limitations section of the discussion now reads:

“Lastly, we were unable to assess differences in vaccine safety by level of deprivation, which should be prioritised as a future area of research.” [lines 447-448]

Presentation of the results

Overall, as shown in Table 2, over 99% of children received the BNT162b2 vaccine as their first and second dose. To me then, the main and most reliable results from this paper pertain to BNT162b2. It would make the paper easier to follow and the tables MUCH easier to interpret in the main paper if results were only shown for the BNT162b2 vaccine in Table 3. Full results for the other two vaccines can be given in the supplementary and discussed briefly in the main paper, but the focus should be on BNT162b2, since that is where the overwhelming amount of data is.

RESPONSE: Thank you for this suggestion. While the results for BNT162b2 are the most robust due to the large sample size, we feel that it is important to report the results for all our pre-specified analyses in the main paper (including all vaccine exposures) for full transparency. However, we have simplified Table 2 to report the characteristics of children who received at least one dose of any COVID-19 vaccine, and only report characteristics by uptake of each vaccine type in Supplementary Table 1.

Staying on Table 3, I think it would also be more transparent if all adverse outcomes were given for each subsection and “0” entered where there were no observed events. At the moment it is difficult to compare between age groups and conditions because each section has different outcomes. This would include a set of “0”s for SARS-CoV-2 vaccination in 5-11s.

RESPONSE: Thank you for this suggestion. We have updated Table 3 to include all the outcomes for each age group and exposure.

Figures 1 and 2: I think a little more explanation in the caption would be helpful (e.g. be explicit on comparison to baseline). Figure 1 seems to have 3 colour dots (blue, green, red) but the legend only

has 2 colours (blue and red). Be clear in the caption that absence of a dot is absence of events (I assume> e.g no unvax dot for epilepsy in women in 5-11s).

RESPONSE: Thank you for this suggestion. We have added more details to the legends for Figures 1 and 2 along the lines helpfully suggested by the Reviewer.

The legend for Figure 1 now reads: ***“Figure 1. Risk of serious outcomes following COVID-19 vaccination with the BNT162b2 vaccine or SARS-CoV-2 infection before and after vaccination in adolescents aged 12-17 years. Estimated number of excess events per million (95% CI) based on incidence rate ratios of each outcome in the 1-42 days following a SARS-CoV-2 positive test (before or after vaccination) compared to baseline period are presented where there were at least five events during the exposure period and when number of excess events is greater than zero. MIS-C = multisystem inflammatory syndrome; ITP = idiopathic or immune thrombocytopenic purpura; ADEM = acute disseminated encephalomyelitis”***

The legend for Figure 2 now reads: ***“Figure 2. Risk of serious outcomes following SARS-CoV-2 infection in children aged 5-11 years who had not been vaccinated against COVID-19 prior to infection. Estimated number of excess events per million (95% CI) based on incidence rate ratios of each outcome in the 1-42 days following a SARS-CoV-2 positive test compared to baseline period are presented. Results for ITP and Guillain-Barre Syndrome are not presented as there were fewer than five events during the exposure period. MIS-C = multisystem inflammatory syndrome; ITP = idiopathic or immune thrombocytopenic purpura; ADEM = acute disseminated encephalomyelitis.”***

Figure 2 is lovely, but it would help to have gridlines, it's very hard to match the data on the right hand panel to the respective adverse outcome. Also I am not sure about the title which does not match the caption – it suggests it's just for infection but I think this figure shows after infection AND after each of the 3 possible doses (regardless of infection).

RESPONSE: Thank you for this suggestion. We have added gridlines to Figures 1 and 2 to aid interpretation of the graphs and changed the title for Figure 1 (which was previously Figure 2), which now reads:

“Figure 1. Risk of serious outcomes following COVID-19 vaccination with the BNT162b2 vaccine or SARS-CoV-2 infection before and after vaccination in adolescents aged 12-17 years.”

Supplementary Table 1 is a bit confusing. If this is the *Excluded* cohort surely it's people who did not receive a vaccine dose AND had no recorded SARS-CoV-2 infection? The title and the columns suggest it's an “or”. This does not seem to match the stated study cohort (EITHER SARS-CoV-2 vaccine OR SARS-CoV-2 confirmed infection).

RESPONSE: Thank you for pointing this out. We have updated this table (now Supplementary Table 2) to only include young people who did not receive a COVID-19 vaccine and did not have a SARS-CoV-2 test recorded during the study period. The Results section now reads:

*“The characteristics of the population excluded from the self-controlled case series analysis (i.e. those who did not receive any COVID-19 vaccine or have a SARS-CoV-2 test recorded during the study period) are presented in **Supplementary Table 2.**”* [lines 144-146]

MINOR COMMENTS

- In the introduction (lines 55-56), the authors discuss vaccine coverage of 12-15 year olds but in the results use 12-17 year olds as the age cohort. This is fine, but then I think the introduction should discuss coverage in 12-17 year olds also (acknowledging that the vaccine offer and timing varied between 12-15 and 16-17 year olds – 16/17 year olds had access about 6 weeks earlier than 12-15 year olds).

RESPONSE: We have added the COVID-19 uptake statistics for 16-17-year-olds in the Introduction, which now reads:

“Despite uptake being very high in adults, with over 80% receiving at least one dose of COVID-19 vaccine as of 11th May 2023, uptake has been lower in children, with 62% of 16-17-year-olds, 46% of 12-15-year-olds and 10% of 5-11-year-olds being vaccinated against COVID-19².” [lines 60-63]

- The authors say that there have been few studies on the safety of SARS-CoV-2 vaccines in 5-11 year olds at a large scale, but cite the systematic review by Watanabe et al. Assessment of Efficacy and Safety of mRNA COVID-19 Vaccines in Children Aged 5 to 11 Years: A Systematic Review and Meta-Analysis. JAMA Pediatrics 2023, doi:10.1001/jamapediatrics.2022.6243 (ref 33) which includes data on over 10 million vaccinated children. I think it's fair to say that the safety of vaccination in 5-11 yr olds is pretty well established.

RESPONSE: Thank you for this comment. While the systematic review by Watanabe et al. included over 10 million children across 17 studies, only two of these studies reported the incidence of adverse events following COVID-19 vaccination compared to unvaccinated controls. These studies were both randomised controlled trials that included just over 6000 children (Walter et al., 2022, Creech et al., 2022). We have changed the wording of the Introduction to reflect this, which now reads:

“No serious safety concerns have yet been raised in younger children; however, population-based studies assessing the risk of adverse events following vaccination compared with an unvaccinated group are lacking³³⁻³⁵.” [lines 94-96]

- Results and discussion – while the data are available in the table, it would be good to compare risks of adverse outcomes explicitly between SARS-CoV-2 infection and vaccination, especially for myocarditis (given that this comparison is made in the paper for epilepsy).

RESPONSE: Thank you for this comment. Throughout the paper we have avoided making explicit comparisons between the risk of adverse outcomes following vaccination and SARS-CoV-2 infection. Due to the nature of SARS-CoV-2 testing and reporting, and the fact that we only considered the first recorded positive SARS-CoV-2 test for the analysis of risk of safety outcomes following SARS-CoV-2 infection, the denominator for this analysis will be biased by COVID-19 testing strategy and will not be a complete representation of all COVID-19 infections that occurred in children in the UK. However, the analysis of risk of safety outcomes following COVID-19 vaccination included all children vaccinated against COVID-19, and therefore is representative of the English population of under-18s. We have removed the comparison of risk of hospitalisation with epilepsy following vaccination and SARS-CoV-2 infection in the discussion and added the limitation of unascertained SARS-CoV-2 infections to the discussion, which now reads:

“We were also unable to account for unascertained SARS-CoV-2 infections, therefore, some of the adverse events could be misclassified as being associated with vaccination rather than a SARS-CoV-2 infection that was not recorded and the analysis of adverse outcomes following SARS-CoV-2 infection may have been biased by incomplete COVID-19 testing in the English population.” [lines 431-435]

Also, as pointed out by the Reviewer, the results are available for comparison in the Tables and Figures.

- Lines 248-249: I found this sentence quite hard to parse and it wasn't immediately obvious to me that the authors were saying that rates of myocarditis were substantially higher in 18-24 year olds. I think because I read “in the comparison groups” as “vs the comparison group” but perhaps be explicit and say rates were higher in 18-24 than younger age groups.

RESPONSE: Thank you for highlighting this. We have changed the wording of this sentence which now reads:

“As expected, we observed a substantially increased risk of myocarditis in the 1-42 days following almost all doses of both mRNA vaccines in young adults aged 18-24 years compared to the baseline period, consistent with other studies reporting an increased risk of myocarditis in young adult males following a second mRNA-1273 vaccine dose²⁴⁻²⁶.” [lines 310-313]

- Final sentence: “This study highlights that the decision to vaccinate children and young people with COVID-19 vaccines should be carefully balanced with the risk of infection and complications following SARS-CoV-2 infection, as well as the broader impact of the COVID-19 pandemic on the well-being of young people.” I'm not sure I think this sentence follows from this paper! All decisions to introduce a vaccine (whether for adults or children) require a careful risk-benefit analysis. And countries did carry those out. All the evidence since vaccination of children started is that adverse vaccine events are low and the risk of SARS-CoV-2 infection outweighs that of vaccination. This study adds to that evidence. There is no data in the paper on the wider impact of the pandemic (and are the authors thinking of things like long covid? Or home school? Or something else?) So I'm not sure what point the authors are making here?

RESPONSE: Thank you for highlighting this. We have now removed this final sentence of the Discussion section. The conclusion now reads:

“In summary, we found no strong evidence for increased risks of 11 pre-specified vaccine safety outcomes following COVID-19 vaccination in children aged 5-11 years and no new significant safety concerns in 12-17-year-olds following vaccination with mRNA vaccines recommended for use in these age groups in the UK by the JCVI. Additionally, in unvaccinated children we found increased risks of hospitalisation from seven adverse outcomes including MIS-C and myocarditis following SARS-CoV-2 infection that were either not observed, or were reduced, following vaccination. Overall, our findings support a favourable safety profile of COVID-19 vaccination using mRNA vaccines in children and young people aged 5-17 years.” [lines 447-456]

- Table 3 has some cell entries saying “>=10” (e.g. ADEM) – why? There is nowhere in the methods that provides a reason for that? should it be a “<” sign?

RESPONSE: This was used to prevent disclosing information on one individual in the paper, however, Table 3 has now been updated.

Reviewer #2 (Remarks to the Author):

I would like to thank this opportunity to review the draft entitled "Risks and benefits of BNT162b2, mRNA-1273 and ChAdOX1 vaccines in 5.1 million children and young people". The authors examined the association between COVID-19 vaccination and serious adverse outcomes in the UK nationwide linked data, using the self-controlled case series (SCCS) method. The draft is clearly written, whereas I have concerns in the current study.

Major point:

My strong suggestion is that the authors should conduct not only the SCCS analysis but also a traditional cohort analysis to report absolute incidence rates of the outcomes in children with and without vaccination, as well as their crude and adjusted incidence rate ratios.

I agree that the SCCS is generally more suitable for examining a causal relationship between an exposure such as vaccination and outcomes than the traditional cohort design. However, I have strong concerns whether the results of SCCS alone are used for discussing the risk-benefit balance of vaccination in the underlying population.

The SCCS uses only the data of patients with studied outcomes. The estimands of cohort studies and SCCS seem to be different; the former refers to the effect of the treatment in a population compared to a control group, and the latter to the within-subject effect (S. Greenland. A unified approach to the analysis of case-distribution (case-only) studies Stat Med, 18 (1) (1999), pp. 1-15). Although the authors estimated the excess events per million for each outcome based on the SCCS results, I am not certain whether this is equal to the (adjusted) excess risk comparing vaccinated and unvaccinated children in the underlying population, if the estimands are different between the SCCS and cohort analyses. While I read the reference 58 the authors cited (Wilson, K. & Hawken, S. Drug safety studies and measures of effect using the self controlled case series design. Pharmacoepidemiol Drug Saf 22, 108-110 (2013)), I was not sure whether this important point was mentioned and accounted for in their calculation of absolute risk in the cited paper.

Also, the SCCS is not perfect: the SCCS has its own limitations and requires strong assumptions which may not be true. In particular, I wonder if children with the outcome of interest have the same probability of receiving vaccination, even after 28 days (as a "pre-risk" period) which the authors excluded from the analysis. i.e. it is possible that children with the outcome of interest, which may be "serious", are far unlikely to receive the COVID-19 vaccination later. A cohort study is unlikely to suffer from this bias, and therefore, anyway, the results of cohort study should be shown as part of the "triangulation" approach in epidemiology.

RESPONSE: Thank you for this feedback. As suggested by the Reviewer, we have now conducted an additional traditional cohort study analysis on a subset of the study population that were included in the QResearch database of primary care records. We undertook a matched cohort design, where we matched vaccinated children to unvaccinated children on age, sex and calendar date, to compare the risk of safety outcomes following vaccination relative to an unvaccinated comparator group. We detail the methodology, results and interpretation of the cohort study in the revised paper.

The Methods section now reads:

"Matched cohort analysis

We also conducted a post hoc matched cohort study using the QResearch database of primary care records, linked to hospital episode statistics, COVID-19 vaccination and SARS-CoV-2 infection data. The study population included all vaccinated children with GP records in the QResearch database,

and matched unvaccinated children, irrespective of SARS-CoV-2 test status during the study period. We matched vaccinated children to children of the same age and sex who were unvaccinated at the time that the vaccinated child received their first dose (rounded to the nearest 7 days) at a ratio of 1:1. Unvaccinated people were sampled from the whole cohort and included people who were vaccinated later in the study period, and these people were censored on date of their first vaccination. The same matched pairs were included in the analysis of the second and third doses for those who received these doses, and where the matched unvaccinated person remained unvaccinated at the time of the second and third doses being received by the vaccinated person.

We estimated incidence rates and fitted conditional Poisson regression models to estimate IRRs of each outcome in the 1-42 days following a first, second or third dose of BNT162b2, mRNA-1273 or ChAdOX1 COVID-19 vaccine. We estimated unadjusted IRRs and IRRs adjusted for self-reported ethnicity (white, non-white, missing), quintile of deprivation (based on Townsend score) and presence of comorbidity (yes/no) that would result in inclusion in clinical risk group (diagnosis prior to vaccine being available).

We additionally conducted a matched cohort analysis in 18-24-year-olds for comparison.” [lines 538-557]

The Results section now reads:

“Matched cohort study

Our matched cohort analysis included 1,490,219 children aged 5-11 years and 1,409,745 adolescents aged 12-17 years. Characteristics of the cohort are detailed in **Supplementary Table 8**.

Incidence rates of vaccine safety outcomes in the 1-42 days following each vaccine dose and following SARS-CoV-2 infection in vaccinated and unvaccinated children are presented in **Supplementary Table 9**. Incidence rates for all outcomes were significantly higher following SARS-CoV-2 infection compared to COVID-19 vaccination.

We matched 160,262 children aged 5-11 years and 848,186 adolescents aged 12-17 years who had received at least one dose of COVID-19 vaccine to a child of the same age and sex who had not received any COVID-19 vaccine doses by the date of the vaccinated child’s first vaccine dose (characteristics of matched cohort reported in **Supplementary Table 10**). We estimated both unadjusted IRRs and IRRs adjusted for self-reported ethnicity (white, non-white, missing), quintile of deprivation (based on Townsend score) and presence of comorbidity (yes/no), which are presented in **Supplementary Table 11**.

As in the self-controlled case series analysis, we identified an increased risk of hospitalisation with epilepsy in the 1-42 days following a second dose of COVID-19 vaccine with BNT162b2 in 12-17-year-olds (unadjusted IRR 1.77, 95%CI 1.05-2.99, adjusted IRR 3.88, 95%CI 1.27-11.86), but did not find significantly increased risks of appendicitis or myocarditis with BNT162b2 vaccination in adolescents (**Supplementary Table 11**).

We identified additional increased risks of anaphylaxis and appendicitis in 12-17-year-olds following a first dose of BNT162b2 (unadjusted IRR 3.71, 95%CI 1.23-11.14 and unadjusted IRR 1.37, 95%CI 1.05-1.80, respectively) and an increased risk of hospitalisation with epilepsy following a first dose with BNT162b2 in 5-11-year-olds, although the confidence interval was very wide reflecting the uncertainty of the estimate (unadjusted IRR 16.00, 95%CI 2.12-120.65) (**Supplementary Table 11**).

In general, the estimates from the matched cohort study were in agreement with the results from the self-controlled case series analysis in under-18s.” [lines 259-286]

The Discussion section now reads:

“A substantially increased risk of appendicitis was observed in adolescents following a second dose of ChAdOX1, with an additional 512 (95%CI 283-599) cases expected per million. This estimate is based on a small sample size as the ChAdOX1 vaccine was not approved for use in under-40s in the UK from April 2021^{3,53}. Additionally, the increased risk was only observed after four weeks following the date of vaccination, and was not identified in the matched cohort study, suggesting the evidence from this study for a causal association between appendicitis and ChAdOX1 vaccination is weak.” [lines 352-358]

“We did not identify an increased risk of hospitalisation with epilepsy following vaccination with ChAdOX1 in the matched cohort study. Our findings, together with a recent study that found evidence for increased risk of cardiac death in young women following a first dose of non-mRNA vaccine⁵⁷, suggest that further work would need to be done to ensure the safety of ChAdOX1 in young people if it were to be used in future vaccination programmes.” [lines 375-380]

“In the matched cohort study including >160,000 vaccinated 5-11-year-olds we additionally found an increased risk of hospital admission with epilepsy following a first dose with BNT162b2 compared to unvaccinated children. However, given that epilepsy was not identified as a safety signal in this age group in the self-controlled case series analysis and the lengthy diagnosis pathway for epilepsy as described above, this potential increased risk is most likely restricted to children with pre-existing epilepsy.” [lines 386-392]

“We also assessed the robustness of our results through several sensitivity analyses, a matched cohort analysis and a parallel analysis in 18-24-year-olds to ensure that the results from our study were consistent with the current evidence base in adults.” [lines 418-421]

Minor point:

Lines 158-160 ("mRNA-1273 vaccine We found no evidence for significantly increased risks for any of the pre-specified outcomes in the 1-42 days following a first, second or third dose of mRNA-1273 vaccine in 12-17-year-olds.") could mislead some readers (who are not familiar with epidemiology) as if mRNA-1273 is safer than BNT162b2 vaccine. As the authors mention elsewhere in the draft, the detection of "statistical significance" largely depends on the number of children receiving the vaccine. The authors should be careful about this point when summarizing the results, and should not mislead some readers.

RESPONSE: Thank you for this comment. We have changed the text to highlight the sample size considerations that should be considered in the interpretation of the results. The Results section now reads:

“We found no evidence for significantly increased risks for any of the pre-specified outcomes in the 1-42 days following a first, second or third dose of mRNA-1273 vaccine in 12-17-year-olds (Table 3b). However, this analysis lacked power to detect statistically significant associations, except for very large effect sizes, as less than 0.1% of adolescents received a first or second dose of mRNA-1273 vaccine.” [lines 189-193]

Reviewer #3 (Remarks to the Author):

General comments

In this article, the authors assessed a set of risks associated either to COVID-19 vaccination or SARS-CoV-2 infection in children and adolescents in the UK. Using a self-controlled design, they find that increased risks after infection - notably of myocarditis and MIS-C - outweigh the risks after vaccination. The question of the balance of risk and benefit of Covid vaccination in children and adolescents is very important and more debatable than that in adults.

I see two major concerns about the study findings.

1. Imperfect detection of SARS-CoV-2 infection

- Whereas vaccine exposure ought to be precisely measured, infection events are typically under reported, in that only a fraction of infections results in recorded confirmation tests. And this is likely differential with greater chance to confirm an infection in those with a severe and hospitalised outcome than with mild or asymptomatic forms. This would overestimate the risk after infection and the relative benefit of vaccination.

- The authors should better address this limitation and particularly discuss the relevance of the "million exposed" denominator of infected used in absolute risk calculation that provides support to their principal conclusion.

RESPONSE: Thank you for this comment. We have addressed the limitation of unascertained SARS-CoV-2 infections in response to a previous comment from Reviewer 1. Briefly, as we used routinely collected data, unascertained SARS-CoV-2 infections are a limitation of this analysis as highlighted by the Reviewer. The outcomes of the study were hospitalisations with a pre-specified safety outcome that occurred in the 1-42 days following vaccination or SARS-CoV-2 infection but does not include outcomes that occurred on the day of vaccination, or the day of the positive SARS-CoV-2 test being reported, therefore limiting the likelihood of differential testing related to the occurrence of the outcome. Unascertained infections could also lead to safety outcomes being misclassified as being associated with vaccination rather than a SARS-CoV-2 infection that was not recorded, which will be particularly relevant to outcomes that are known complications of COVID-19 infection, such as MIS-C. We have included this limitation in the Discussion section which now states:

"We were also unable to account for unascertained SARS-CoV-2 infections, therefore, some of the adverse events could be misclassified as being associated with vaccination rather than a SARS-CoV-2 infection that was not recorded and the analysis of adverse outcomes following SARS-CoV-2 infection may have been biased by incomplete COVID-19 testing in the English population." [lines 431-435]

With regards to the "million exposed" denominator, this was used to put the results into the population context, as the risks associated with COVID-19 vaccines and infections are relevant at the population level. Given that 5.1 million children and adolescents who had received at least one COVID-19 vaccine or had a recorded COVID-19 infection were included in the main analysis, the "million exposed" denominator was we believe appropriate for the absolute risk calculations.

2. Time period of infections

- There is a major difference in symptoms occurrence between variants and the authors may have assessed serious outcomes post infection that are specific to variants of the first part of the study period.

- The effect of infection either before or after vaccination is confounded by the difference in pathogenicity between variants. See for example <https://doi.org/10.1093/cid/ciac553> where post

infection risk of MIS-C is found lower with delta and omicron variants, even pre vaccination.

- This ought to be addressed to interpret the results.

RESPONSE: Thank you for this comment and for providing this useful reference. We have now addressed this limitation in the Discussion where we have now included the suggested reference:

“Fourth, we were unable to determine the effect of SARS-CoV-2 variant on the risk of adverse events, as detailed data on the viral variant of SARS-CoV-2 underlying recorded infections was also not available in our database. Fifth, although we adjusted for seasonal effects in the self-controlled case series models, we did not explicitly investigate the effect of different SARS-CoV-2 variants or the time period during which the infection occurred on the risk of adverse events. For example, the incidence of MIS-C has been reported to be lower during the periods when Delta and Omicron variants were dominant, even before 12-15-year-olds started being vaccinated, compared to the period when the Alpha variant was dominant⁵⁸. Given that >98% of children in this study were vaccinated against COVID-19 during the periods that Delta and Omicron were dominant, we may expect a lower incidence of MIS-C following infection in vaccinated children, who were unlikely to have been infected with the Alpha variant, compared to unvaccinated children, who had a higher likelihood of being infected with the Alpha variant.” [lines 435-446]

A few minor points that could be considered to improve the paper:

- line 210: It may be helpful to remind the readers that the positive control outcome is anaphylaxis. And I cannot confirm from the tables that "the estimates agreed with the main results".

RESPONSE: We have included additional detail on how we assessed the robustness of the results in the Results section, including our analysis of the positive control outcome of anaphylaxis, which now reads:

“Robustness of the self-controlled case series results

The robustness of the results of the self-controlled case series analyses were assessed by (1) checking that the risk of outcomes during the pre-vaccination period (month prior to vaccination to account for potential bias of people with recent hospitalisation being less likely to get vaccinated) was lower than the baseline period and (2) checking that the risk of the positive control outcome (anaphylaxis) was higher than the baseline period. In the vast majority of analyses the estimates of the pre-vaccination period agreed with what was expected and the risk of anaphylaxis following vaccination or SARS-CoV-2 was consistently higher across all analyses (data not shown).” [lines 249-257]

- line 237: I don't get this last part of the sentence. The sample of vaccinated children included in this study is either insufficient to observe any potential events (the authors do not seem to state that) or sufficient - which indicates no safety issue in the entire population. The authors should clarify the role of the uptake rate.

RESPONSE: We have clarified that the number of young children receiving ChAdOX1 and mRNA-1273 vaccines, and the number of adolescents receiving a first or second dose of mRNA-123 vaccine, was too low to detect safety signals, by adding the following to the Results section:

“In children aged 5-11 years, we did not observe an increased risk of any of the pre-specified outcomes in the 1-42 days following any dose of BNT162b2, mRNA-1273 or ChAdOX1 COVID-19 vaccine (Table 3a). However, given that less than 0.1% of vaccinated 5-11-year-olds received a

ChAdOX1 or mRNA-1273 vaccine, the probability of type II errors was high as the sample size was too small to detect statistically significant associations for these vaccines.” [lines 150-154]

“We found no evidence for significantly increased risks for any of the pre-specified outcomes in the 1-42 days following a first, second or third dose of mRNA-1273 vaccine in 12-17-year-olds (Table 3b). However, this analysis lacked power to detect statistically significant associations, except for very large effect sizes, as less than 0.1% of adolescents received a first or second dose of mRNA-1273 vaccine.” [lines 189-193]

- line 273-275: This sentence is unclear and needs rewriting

RESPONSE: The sentence has been rewritten to make clearer. It now reads:

“However, a diagnosis of epilepsy is made over a period of time as it typically involves outpatient referral, a magnetic resonance imaging (MRI) scan and an electroencephalogram⁴⁷. Therefore, this reported increased risk of epilepsy is highly unlikely to reflect new-onset epilepsy triggered by the vaccine.” [lines 318-321]

- line 340: I would tone down this statement on "sufficient power". It seems surprising not to retrieve the association between mRNA-1273 vaccine and myocarditis in adolescents that was found in many settings.

RESPONSE: Thank you for this comment. We have revised the strengths section of the Discussion, which now reads:

“Second, the large sample size allowed us to investigate rare outcomes, particularly following vaccination with BNT162b2, which could not be assessed through clinical trials.” [lines 414-416]

- line 372: The authors should better explain the nature and use of the QResearch database.

RESPONSE: In the original version of the manuscript, the QResearch database was used to supplement the demographic data from the NIMS database of COVID-19 vaccination and the SGSS database of SARS-CoV-2 testing. In the revised version of the manuscript, we have conducted a matched cohort study as suggested by Reviewer 2, which is based on the QResearch primary care database. We have added a full description of the matched cohort study to the Methods, as described in our response to Reviewer 2’s comment. The description of the data used in the matched cohort study reads:

“We also conducted a post hoc matched cohort study using the QResearch database of primary care records, linked to hospital episode statistics, COVID-19 vaccination and SARS-CoV-2 infection data. The study population included all vaccinated children with GP records in the QResearch database, and matched unvaccinated children, irrespective of SARS-CoV-2 test status during the study period.” [lines 541-543]

Using data from the QResearch database, we were able to adjust the models for ethnicity, level of deprivation (based on Townsend score) and co-morbidities recorded in each child’s GP record, which is also described in the Methods:

“We estimated unadjusted IRRs and IRRs adjusted for self-reported ethnicity (white, non-white, missing), quintile of deprivation (based on Townsend score) and presence of comorbidity (yes/no) that would result in inclusion in clinical risk group (diagnosis prior to vaccine being available).” [lines 553-556]

- line 387: The authors should fully explain the selection process for the studied outcomes.

RESPONSE: We have clarified that we selected outcomes *a priori* through a combination of literature reviews and neurological, cardiology, paediatric and immunology/vaccine expertise in the Methods section. We have also highlighted that the selected outcomes are in line with other studies exploring COVID-19 vaccine safety in young people. The Methods now read:

"We a priori selected severe outcomes resulting in hospital admission or death which are monitored by national medical regulatory authorities, clinical trials, post-marketing surveillance, emerging scientific literature and neurological, cardiology, paediatric and immunology/vaccine expertise available. Our pre-defined outcomes were previously reported COVID-19 infection- and COVID-19 vaccination-related adverse events with strong evidence in young people: myocarditis^{21,22}, MIS-C³⁸ and myositis⁴⁵ and those in adults: Guillain-Barre syndrome⁴³, demyelinating disease⁶, ITP³⁹ and appendicitis⁴⁴ as well as adverse events reported following any vaccination during childhood and early young adulthood: epilepsy⁴⁰, acute pancreatitis⁴¹, ADEM⁴² and angioedema⁴⁶. The selected outcomes are consistent with previous studies reporting COVID-19 vaccine safety in young people⁵⁹. Anaphylaxis was also included as a positive control outcome because it can occur shortly after vaccination⁴⁶." [lines 483-493]

- line 399: How was handled the overlap of risk periods between dose 1 and 2 within 42 days in the SCCS method ?

RESPONSE: We have clarified this in the Methods section, which now reads:

"In the case where two vaccine doses were given within 42 days of each other, the risk period of the earlier dose was truncated on the date prior to the day that the next vaccine dose was received." [lines 507-509]

- line 411-426: Did the SCCS model account for the violation of the assumption that the occurrence of an event does not influence subsequent exposure ? I would imagine that children developing one of the outcome after infection or previous vaccine dose would not receive a subsequent dose as planned.

RESPONSE: We included a separate "pre-risk" exposure period of 1-28 days before each vaccine exposure to account for the fact that children who have had a recent hospital admission, including due to an adverse event, are likely to delay vaccination. This is specified in the Methods section:

"People with a recent hospital admission may delay vaccination, therefore a pre-risk period of 1-28 days before each exposure was excluded from the baseline period to account for this potential bias³⁶." [lines 509-511]

- Table 5 has no column headings.

RESPONSE: Thank you for highlighting this. The column headings have now been added to Table 5.

REVIEWERS' COMMENTS

Reviewer #1 (Remarks to the Author):

Thank you for very much for the detailed response to my comments which have all been addressed to my satisfaction.

I look forward to seeing this important paper in print!

Reviewer #2 (Remarks to the Author):

Thank you very much for your efforts and sufficient responses to my comments.

Reviewer #3 (Remarks to the Author):

The authors have successfully addressed the points of my initial review. The paper has been generally improved.

I have no further comments and recommend publication.

Reviewer #3 (Remarks on code availability):

The github repository contains STATA scripts relevant to the self-controlled case series method used in several former papers from the team.

I cannot reproduce the analysis as I am not a STATA user. But I doubt the specific analyses of this paper could be reproduced as the repository has not been updated for 2 years.

There is no way for instance to assess the way the cohort study was processed and analyzed in this paper.

My conclusion is that the code for this paper is not entirely available.

Safety outcomes following vaccination with BNT162b2, mRNA-1273 and ChAdOX1 COVID-19 vaccines and SARS-CoV-2 infection in 5.1 million children

Emma Copland, Martina Patone, Defne Saatci, Lahiru Handunnetthi, Jennifer Hirst, David P J Hunt, Nicholas L Mills, Paul Moss, Aziz Sheikh, Carol AC Coupland, Anthony Harnden, Chris Robertson, Julia Hippisley-Cox

Response to reviewer comments

Reviewer #1 (Remarks to the Author):

Thank you for very much for the detailed response to my comments which have all been addressed to my satisfaction.

I look forward to seeing this important paper in print!

Thank you for this comment.

Reviewer #2 (Remarks to the Author):

Thank you very much for your efforts and sufficient responses to my comments.

Thank you for this comment.

Reviewer #3 (Remarks to the Author):

The authors have successfully addressed the points of my initial review. The paper has been generally improved.

I have no further comments and recommend publication.

Thank you for this comment.

Reviewer #3 (Remarks on code availability):

The github repository contains STATA scripts relevant to the self-controlled case series method used in several former papers from the team.

I cannot reproduce the analysis as I am not a STATA user. But I doubt the specific analyses of this paper could be reproduced as the repository has not been updated for 2 years.

There is no way for instance to assess the way the cohort study was processed and analyzed in this paper.

My conclusion is that the code for this paper is not entirely available.

Thank you for this comment. We have updated the Code Availability statement to clarify the code that is available, which now reads:

“A sample of the code used for a similar study has been deposited in the public git repository of the research group, available at <https://github.com/qresearchcode/COVID-19-vaccine-safety>. This sample code can be used to run a self-controlled case series analysis in STATA.” [lines 579-582]